



**Impact of Athabasca oil sands operations on mercury levels in air and deposition**
Ashu Dastoor[1], Andrei Ryjkov[1], Gregor Kos[2], Junhua Zhang[3], Jane Kirk[4], Matthew Parsons[5] and
Alexandra Steffen[3]
[1]Air Quality Research Division, Environment and Climate Change Canada, 2121 Trans-Canada
Highway, Dorval, Québec, Canada
[2]Department of Chemistry and Biochemistry, Concordia University, 7141 Sherbrooke Street
West, Montreal, Québec, Canada
[3]Air Quality Research Division, Environment and Climate Change Canada, 4905 Dufferin Street,
Toronto, Ontario, Canada
[4]Aquatic Contaminants Research Division, Environment and Climate Change Canada, 867
Lakeshore Road, Burlington, Ontario, Canada
[5] Meteorological Service of Canada, Environment and Climate Change Canada, 9250 49 Street
NW, Edmonton, Alberta, Canada
**Correspondence**: Ashu Dastoor (ashu.dastoor@canada.ca)
**Abstract**
Oil sands upgrading facilities in the Athabasca Oil Sands Region (AOSR) in Alberta, Canada, have
been reporting mercury (Hg) emissions to public government databases (National Pollutant
Release Inventory (NPRI)) since the year 2000, yet the relative contribution of these emissions to
ambient Hg deposition remains unknown. A 3D process-based global Hg model, GEM-MACH-
Hg, was applied to simulate the Hg burden in and around the AOSR using NPRI reported oil sands
Hg emissions from 2012 (59 kg) to 2015 (25 kg) and other regional and global Hg emissions. The
impact of oil sands emissions (OSE) on Hg levels in the AOSR, relative to contributions from
sources such as global anthropogenic and biomass burning emissions (BBE), was assessed. In
addition, the relative importance of year-to-year changes in Hg emissions from the above sources
and meteorological conditions to inter-annual variations in Hg deposition was examined. Model
simulated surface air concentrations of Hg species and annually accumulated Hg in snowpacks
were found comparable to independently obtained measurements in the AOSR, suggesting
consistency between reported Hg emissions from oil sands activities and Hg levels in the region.



As a result of global-scale transport of gaseous elemental Hg (Hg(0)), surface air concentrations
of Hg(0) in the AOSR reflected the background Hg(0) levels in Canada (1.4 ng m$^{-3}$, AOSR;
1.2-1.6 ng m$^{-3}$, Canada) with negligible impact from OSE. Highly spatiotemporally variable
wildfire Hg emission events led to episodes of high ambient Hg(0) air concentrations of up to 2.5
ng m$^{-3}$ during the burning season. By comparison, average air concentrations of total oxidised Hg
(gaseous plus particulate; efficiently deposited Hg species) in the AOSR were elevated by 60%
above background levels (2012-2013) within 50 km of the oil sands major upgraders as a result of
OSE. Annual average Hg deposition fluxes in the AOSR were within the range of the deposition
fluxes measured for the entire province of Alberta (15.6-18.3 µg m$^{-2}$y$^{-1}$, AOSR (2012-2015); ~14-
25 µg m$^{-2}$y$^{-1}$, Alberta (2015)).  Winter (November-April) and summer (June-August), respectively,
accounted for 20% and 50% of the annual Hg deposition in the AOSR. On a broad spatial scale,
imported Hg from global sources dominated the annual Hg deposition in the AOSR, with present-
day global anthropogenic emissions contributing to 40% (< 1% from Canada excluding OSE), and
geogenic emissions and re-emissions of legacy mercury deposition contributing to 60% of the
background Hg deposition. Further, wildfire events contributed to regional Hg deposition with
enhancements of 1-13% across 200 km range of major oil sands sources. In contrast, oil sands Hg
emissions were responsible for significant Hg deposition enhancements in the immediate vicinity
of oil sands Hg emission sources, up to 100 km in winter and up to 30 km in summer. Hg deposition
enhancements related to oil sands emissions were about 10 times larger in winter than summer
(average enhancement of 250 – 350% in winter and ~35% in summer within 10 km of OSE, 2012-
2013). In addition, snowpack Hg loadings and wintertime Hg deposition displayed significantly
higher inter-annual variations compared to summertime deposition due to changes in
meteorological conditions (such as precipitation amounts, wind speed, surface air temperature,
solar insolation, and snowpack dynamics) as well as oil sands emissions. For example, a large
snowmelt event at the end of February in 2015 effectively removed about half of the accumulated
mercury in snow, contributing to (observed and modeled) low annual snow Hg loadings. Inter-
annual variations in meteorological conditions were found to both exacerbate and diminish the
impacts of OSE on Hg deposition in the AOSR, which can confound the interpretation of trends
in short-term environmental Hg monitoring data. In winter, within 10 km of major oil sands
sources, variations in meteorology led to Hg deposition reduction by 17% in 2014 and increase by
10% in 2015 and decline in OSE lowered Hg deposition by 35% (2014) and 56% ( 2015), resulting



in overall reductions in wintertime Hg deposition of 52% (2014) and 46% (2015), relative to 2012.
By comparison, annually, changes in meteorology and BBE in 2014-2015 (relative to 2012) led to
Hg deposition increases of 1-6% and 2%, respectively, and decline in OSE lowered deposition by
15-22%, resulting in overall reduction in Hg deposition of 7-20% within 10 km of oil sands
sources. Hg runoff in spring flood, comprising the majority of annual Hg runoff, is mainly derived
from seasonal snowpack Hg loadings and mobilization of Hg deposited in surface soils, both of
which are sensitive to Hg emissions from oil sands developments in proximity of sources. Model
results suggest that sustained efforts to reduce anthropogenic Hg emissions from both global and
oil sands sources are required to reduce Hg deposition in the AOSR.

### 73    Introduction

Mercury (Hg) is a toxic element that accumulates in fish and mammals near the top of the food
web, including humans (e.g., through consumption of contaminated fish), where it exhibits long-
term toxic effects (UNEP, 2018). Hg is emitted to the atmosphere from geogenic sources such as
volcanoes and the weathering of Hg-containing rocks, anthropogenic sources such as fossil fuel
burning, metal smelting and artisanal gold mining, and through the re-emission of Hg historically
deposited from anthropogenic and natural sources onto soils, surface waters, and vegetation
(UNEP, 2013). Atmospheric Hg exists mainly in three forms: gaseous elemental mercury (Hg(0)
or GEM), gaseous oxidized mercury (gaseous Hg(II); GOM), and particle bound mercury (particle
bound Hg(II); PBM). The sum of GOM and PBM is referred to as total oxidised mercury (TOM)
and the sum of gaseous mercury species (i.e., GEM and GOM) is referred to as total gaseous
mercury (TGM) in this study. GEM/TGM and TOM are better indicators to compare observation
and model estimates of mercury for the purpose of this study, because of speciation uncertainties
associated with the determination of GOM and PBM species (Gustin et al., 2013). Deposition of
atmospheric Hg species by rain and snow (i.e., wet deposition), and by interfacial uptake on
various surfaces such as soils, vegetation, water, and snowpack (i.e., dry deposition) are the
pathways that contribute to Hg loadings in ecosystems. Typically, atmospheric GEM
concentrations are found to be 2-3 orders of magnitude higher (in the low ng m$^{-3}$ range) than GOM
and PBM (typically in the lower pg m$^{-3}$ range) because GEM is the dominant atmospheric Hg
species emitted to air and the reactivity of the latter (GOM and PBM) leads to efficient dry and
wet deposition removal of these species close to sources. Stability and volatility of GEM results



in its long lifetime in the atmosphere, with six months to one year, allowing for transport and
distribution on a global scale, and re-emission from planetary surfaces (UNEP, 2013).

On a global scale, dry deposition of GEM by vegetation-uptake over land and wet deposition of
TOM produced by atmospheric oxidation of GEM are the dominant pathways of Hg removal
(Obrist et al. 2016; Wright et al., 2016; Zhou et al. 2021). Primary emissions of GOM and PBM
from industrial sources are an important contributor to dry and wet depositions of Hg on a local to
regional scale. Once Hg is deposited to surfaces, it can be reduced and re-emitted back as GEM to
the air and, thus, Hg redistributes and accumulates in the aquatic and terrestrial environments
globally. Hg also inhibits enzymatic processes and reacts with organic compounds. This leads to
the formation of toxic, and bioaccumulating, methyl-Hg, primarily in aquatic systems, which is
the principal cause of a severe neurological syndrome known as "Minamata Disease". In order to
reduce the amount of Hg released to the environment and limit its exposure to humans, an
international treaty, the Minamata Convention on Mercury, was adopted in 2017 (UN, 2017).

Anthropogenic emissions of Hg to air from global sources stand at an estimated 2220 t $y^{-1}$ in 2015
(UNEP, 2018). Canadian anthropogenic Hg emissions were estimated at about 4.3 t $y^{-1}$ (less than
0.2% of global anthropogenic emissions) in 2015, with an estimated 58% coming from point
sources such as coal-fired power plants and smelters, and 42% from area sources (Zhang et al.,
2018; see Figure 1). Anthropogenic Hg emissions in Canada have declined by 85% from 1990 to
2010 (from ~ 35 to 5 t $y^{-1}$), with major reductions from sectors such as the non-ferrous metal
mining and smelting (-98%), chemical industries (-95%), waste (-76%), iron and steel industries
(-54%) and electric power generation (-30%) (CMSA, 2016). However, due to the steady increase
in development of the oil sands, the upstream petroleum sector has shown increases in Hg
emissions and accounted for approximately 4.6% of the total Canadian Hg emissions in 2010
(CMSA, 2016).

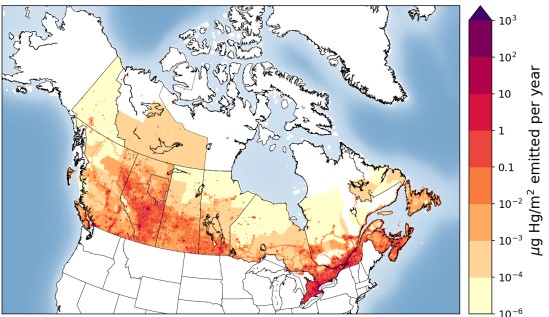


Figure 1: Spatial distribution of anthropogenic Hg emissions in Canada in 2015 (~ 4.3 t/y). The
Athabasca Oil Sands Region is indicated with an approximate rectangular shape within
northeastern Alberta, bordering Saskatchewan.

The Athabasca Oil Sands Region (AOSR) in the northeastern portion of the Canadian province of
Alberta (see Figure 1) is a zone of extensive natural resource development. The large natural
deposits of bitumen, a heavy crude oil, contained in a mixture of water and clay (called "oil sands")
has led to establishment of large-scale mining and upgrading activities in the area north of Fort
McMurray, Alberta (AB) (see map Figure 2). Surface mining and in-situ recovery methods are
used to extract bitumen and then upgrade it to synthetic crude oil (Alexander and Chambers, 2016;
Larter and Head, 2014). Point source emissions of organics and heavy metals, including Hg,
originate from mining activities and upgrading facilities in the AOSR. The upgraders are operated
by the companies Suncor, Syncrude, and Canadian Natural Resources.The upgrading process also
includes the removal of impurities consisting of sulfur and nitrogen-containing compounds by
catalytic hydrotreatment, with volatile hydrogen sulfide and ammonia as by-products. Trace metals
contained in the heavy asphaltene fraction are also removed by either stabilization, rejection, or
upgrading of asphaltenes (Jia, 2014). The yearly amounts of total Hg emissions from Athabasca
oil sands facilities, for the years 2012 to 2015, were between 69 and 25 kg. These annual emissions
exhibited an overall downward trend (for details see Table 1 and Figure 3).






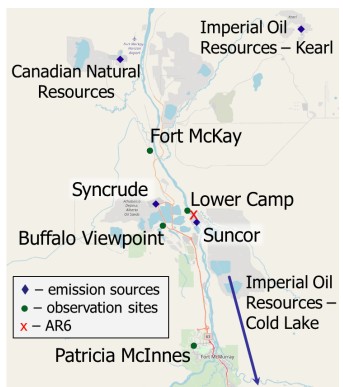

Figure 2: Map of the AOSR with the main point sources for Hg emissions from oil sands developments, and air observation sites. "AR6" marks the approximate midpoint of operations as defined by Kelly et al. (Kelly et al., 2010).

In 2010, Kelly et al., reported increased concentrations of 13 different trace metals including Hg in the surface waters of the Athabasca River and its tributaries in the oil sands region (Kelly et al., 2010). Observed concentrations were higher near oil sands operations than away from the potential sources. Comparison of upstream and downstream data showed consistently higher concentrations for downstream sites. A similar set of observations was made for Hg in surface snow samples, more specifically, Hg bound to particulates. Concentrations in accumulated snow collected near oil sands developments in March averaged 861 ng m$^{-2}$ compared to less than 100 ng m$^{-2}$ for background measurements (Kelly et al., 2010). Several oil sands installations were identified as potential sources for the elevated observations of Hg, but no direct link between sources and observations was established. Specifically, upgraders were discussed as a source for high Hg levels in the region. Other potential sources included fly ash, road dust, land clearing and mining operations.

To address the lack of Hg monitoring and attribution of sources, as concluded by Kelly et al. (2010), several follow-up studies were conducted with the intention to establish a conclusive link between measured pollutant concentrations and potential sources in the AOSR (Kelly et al., 2010; Cooke et al., 2017; Kirk et al., 2014; Emmerton et al., 2018; Lynam et al., 2018; Willis et al., 2019; Willis et al., 2018; Gopalapillai et al., 2019). In addition to water and snow samples, other media have been investigated, such as air, biota and sediments. Cooke et al. (2017) studied lake sediment





cores sampled from 20 lakes at various distances from oil sands operations, including two of them
in the near vicinity (i.e., within 20 km) of the two major upgrading facilities (Suncor and Syncrude)
within the region (or site AR6 as designated by Kelly (2010)) (Cooke et al., 2017). The cores
provided trace metal data for approximately the past 100-250 years.  The cores showed that Hg
concentrations have increased by a factor of 3, reflecting the generally  accepted scientific finding
that global Hg has increased 3-fold as a result of anthropogenic activities since the industrial
revolution. No additional increase of Hg concentrations was detected related to the beginning of
oil sands operations in the late 1960s. This contrasts with Kelly et al. (2010) and the follow-up
study by Kirk et al. (2014) that showed higher Hg loadings in the accumulated snowpack and
surface water sampled closer to the mining and upgrading facilities in the AOSR in early spring
(March), mostly consisting of PBM of atmospheric origin. While Hg levels were closely correlated
with other trace metal concentrations originating from oil sands activities such as nickel and
vanadium, no direct causal link with air emissions of Hg, as reported to the National Pollutant
Release Inventory (NPRI), was established. Gopalapillai and coworkers recently reported temporal
trends in snowpack loadings of total Hg (THg) and methyl mercury (MeHg) (and 44 other
elements) (Gopalapillai et al., 2019). Using a composite of snowpack profile samples collected
between 2011 and 2016 and data from previous campaigns, a decrease in THg loadings from an
average of 510 ng m$^{-3}$ in 2008 to 175 ng m$^{-3}$ in 2016 was found within 8 km from AR6. However,
due to the limited temporal coverage (with measurements for THg starting in 2008), the authors
suggested a need for additional studies to understand the impact of Hg in the AOSR.

A recent study by Emmerton et al. (2018) examined lake water samples and related observed Hg
and methyl-Hg concentrations to local geology, watershed conditions, and to oil sands activities,
with the latter only contributing an estimated <2% of the overall Hg deposited (Emmerton et al.,
2018). Long-range transport and biomass burning (i.e., forest fires) were suggested to be the major
sources of Hg (Emmerton et al., 2018). Similarly, in a recent study of wet deposition data by
Lynam et al. (2018), very low fluxes of Hg deposition were calculated, though the study sites used
(AMS6, the Patricia McInnes observation site shown in Fig. 1) were located further away from
emitters. Results suggested that dry deposition could, instead, be a more important pathway of Hg
removal in the region (Lynam et al., 2018).


In an effort to explain the elevated Hg concentrations found in the snowpack and waters near oil
sands mining and upgrading activities, tailings ponds were studied as a potential source of Hg
emissions related to oil sands activities (Willis et al., 2018). However, the water in these ponds
(i.e., the non-recycled portion of process water used to process mined bitumen) were found to be
an insignificant source of THg and MeHg.

The above-mentioned studies illustrate recent progress in the ongoing effort to examine the link
between observed concentrations and anthropogenic sources of Hg in the AOSR. However, in
addition to local emissions, multiple other sources of mercury emissions impact the region,
especially forest fires and worldwide anthropogenic and geogenic (contemporary and legacy)
emissions that are atmospherically transported into the region. Owing to the much larger emissions
of Hg from worldwide sources, as compared to Canadian sources, and the long lifetime of Hg in
air, imported Hg accounts for the majority of the Hg burden in Canada (CMSA, 2016), rendering
the assessment of the impacts of domestic Hg emissions challenging using measurements alone.
While Cooke et al. (2017) investigated the history of Hg deposition in lake catchments via the
study of sediment cores, only two lakes sampled were close enough (within 20 km) to oil sands
activities, whereas most sites were 20 to >50 km away from the oil sands facilities.

After Hg is emitted to air from oil sands mining and upgrading activities, transport, transformation
and deposition processes determine the distribution and amounts of Hg deposited to environmental
media such as vegetation, soils, and water bodies. 3D process-based predictive atmospheric
composition models include process representations (such as atmospheric transport, chemical
transformations, aerosol particle formation and growth, and wet and dry deposition of gases and
particles) and simulate spatiotemporal distributions of pollutants in air and deposition starting from
emissions (anthropogenic and natural) as inputs. These models provide insight into transport and
transformation pathways of pollutants and causal links between emissions and concentrations
observed in environmental media. Models have been applied to study Hg source attribution on
global and regional scales, answering questions such as how much a specific emission source
contributes to local and regional air concentrations and deposition, and how does the pollutant
burden change as industrial activity and related emissions vary (UNEP, 2008; CMSA, 2016;
UNEP, 2018)? Model processes are typically constrained by evaluating simulated pollutant levels



using observation data from ground-based monitoring networks and research campaigns.
Additionally, aircraft measurement data provide observation data on the vertical scale.

Wildfires are important sources of Hg in Northwestern Canada and climate change is intensifying
their frequency (Fraser et al., 2018). Biomass burning primarily releases legacy Hg previously
deposited to foliage and soils (Friedli et al., 2001; De Simone et al., 2015). Using multivariate data
analysis, Parsons et al. (2013) determined contribution from local sources (i.e., oil sands activities)
to be minimal as compared to total gaseous Hg concentrations in the air in the AOSR; however,
the authors noted significant episodes of regional forest fires impacting the observed Hg
concentrations in the air during the summer months (Parsons et al., 2013).

**Objectives**
Observations of atmospheric Hg in the AOSR are limited to surface air GEM concentrations and
Hg loadings in snow. Summertime wet and dry deposition is not measured. Therefore, measured
estimates of annual Hg deposition in AOSR is currently not possible. Furthermore, a quantification
of the relative importance of different Hg emission sources responsible for Hg loadings in the
AOSR is required to prioritize mitigation actions. The 3D mercury model, Global Environmental
Multiscale - Modelling Air quality and CHemistry – Mercury (GEM-MACH-Hg), was applied to
develop a comprehensive understanding of atmospheric Hg and deposition levels and pathways,
and the role of emissions from Athabasca oil sands activities (particularly from bitumen upgraders)
on the spatiotemporal distribution of Hg deposition in AOSR. This study addresses the following
questions:
1. How do air concentrations and ecosystem loadings of Hg species in AOSR compare to

other regions in Canada?

2. What is the level and geographical extent of the contribution of Athabasca oil sands

emissions on Hg in air and deposition?

3. How does the impact of oil sands development on Hg levels in the region compare with

the impacts of two other major sources of Hg in the region, biomass burning and global

emissions?

4. What controls the inter-annual variability in Hg levels in AOSR?





This is the first study that provides a direct connection between Athabasca oil sands Hg emissions
and deposition of Hg in and around the AOSR. A similar approach using the model GEM-MACH-
Hg was previously applied to the assessment of Hg source apportionment at national and global
scales (CMSA, 2016; AMAP/UNEP, 2013; UNEP, 2018).

**The model and emission inputs**
GEM-MACH-Hg (Dastoor et al., 2015) is the mercury version of Environment and Climate
Change Canada's 3D process-based operational air quality forecast model GEM-MACH (Global
Environmental Multiscale - Modelling Air quality and Chemistry; Makar et al., 2018; Whaley et
al., 2018). GEM-MACH includes emissions of gases and aerosols, and simulates meteorological
processes, aerosol microphysics, tropospheric chemistry and pollutant dry and wet removal
processes from the atmosphere. In addition, GEM-MACH-Hg includes emissions, chemistry and
dry and wet removal processes of three Hg species (GEM, GOM and PBM) (Dastoor and Durnford
2014; Dastoor et al., 2008; Durnford et al., 2012; Fraser et al., 2018; Kos et al., 2013; Zhou et al.
2021). The recent version of GEM-MACH-Hg, previously applied to the investigation of the
importance of biomass burning emissions to the Hg burden in Canada (Fraser et al., 2018) and the
role of vegetation Hg uptake (Zhou et al. 2021), was used in this study. Oxidation of GEM and
gas-particle partitioning of oxidixed Hg species (GOM and PBM) are the main chemical
transformation processes, and dry deposition of GEM, GOM and PBM, and wet deposition of
GOM and PBM are the major removal pathways of Hg in the model. Since observations of
snowpack Hg loadings at the end of the winter season are utilized for model evaluation in this
study, a detailed representation of the air-cryosphere Hg exchange and transformation processes
is important. GEM-MACH-Hg includes a dynamic multilayer air-snowpack–meltwater Hg
parameterization, representing Hg accumulation by precipitation and dry deposition to snowpacks,
vertical diffusion and redox reactions in snowpacks, and re-volatilization and meltwater run-off of
Hg species (Durnford et al., 2012). Geospatially distributed global, regional and local emissions
of Hg species (GEM, GOM and PBM) to air from primary geogenic and anthropogenic sources
and re-emissions of previously deposited Hg (legacy Hg) from terrestrial and oceanic surfaces are
included in the model.





Three geographical domains were utilized for the model simulations in this study: global, North
America (NA) and AOSR. A geospatial resolution of 10 km was chosen for the NA domain and
its boundary conditions were determined by the global simulations conducted at $1^0 \times 1^0$ latitude-
longitude resolution. Model simulations for the AOSR were carried out at a finer geospatial
resolution of 2.5 km for an extended AOSR domain with the approximate midpoint adjacent to the
two largest upgrading facilities (called "AR6") (Kelly et al., 2010) and extending as far north as
Hay River, NT, and as far south as Red Deer, AB; the approximate western and eastern extents of
the domain are marked, respectively, by Grande Prairie, AB and Flin Flon, MB.

Geogenic emissions and re-emissions of legacy Hg in soils and oceans (~4200 t y$^{-1}$) emitted as
GEM were distributed as described in Durnford et al., (2012). Wildfire biomass burning Hg
emissions are represented in the model simulations using the FINN (Fire INventory) fire emissions
products (Wiedinmyer and Friedli, 2007; Wiedinmyer et al., 2011) together with vegetation-
specific emission factors (EFs) as described in Fraser et al. (2018). FINN estimated biomass
burning Hg emissions (emitted as GEM) were ~600 t y$^{-1}$ globally, and 10.8 (2012), 11.4 (2013),
15.5 (2014) and 11.1 (2015) Mg/y in Canada, and 13.4 (2012), 10.5 (2013), 11.4 (2014) and 9.5
(2015) Mg/y in the US.

Contemporary global anthropogenic Hg emissions for 2015 (2224 t y$^{-1}$; subdivided into GEM,
GOM and PBM) developed by the Arctic Monitoring and Assessment Programme (AMAP)
(UNEP, 2018) were incorporated into the model for the global scale simulations. For NA and
AOSR domains, GEM-MACH-Hg includes monthly and diurnally varying anthropogenic Hg
emissions in Canada developed by Zhang et al. (2018), based on the NPRI (NPRI) database (2013)
for the major point sources and the 2010 Air Pollutant Emission Inventory (APEI) for the area
sources. Anthropogenic Hg emissions in the United States included in GEM-MACH-Hg were
based on the 2011 National Emissions Inventory (NEI) (EPA), described in Zhang et al., (2018).
Total anthropogenic emissions of Hg in Canada, the United States and worldwide were 4.3, 47 and
2224 t y$^{-1}$, respectively. The GEM:GOM:PBM ratio in the total anthropogenic Hg emissions was
approximately 70%:23%:7%.





For the oil sands activities related Hg emissions, the model's input consisted only of NPRI-
reported air emissions. Possiblilty of fugitive dust from the disturbed landscape due to oil sands
activities as a source of particulate-bound Hg emissions was noted by Kirk et al. (2014). Cooke et
al. (2017) were unable to detect Hg from dust emissions in lake sediments. Comparison of modeled
and observed Hg levels conducted in this study allowed an assessment of whether NPRI reported
oil sands emissions and area sources (APEI) in AOSR capture Hg emissions in the region
comprehensively or whether there are other yet undetermined important sources of Hg emissions
such as from fugitive dust in the AOSR.

NPRI is a mandatory reporting tool for a wide range of contaminants, including Hg, as prescribed
by the Canadian Environmental Protection Act. Facilities are required to report Hg releases, if total
work hours exceed 20, 000 and if a reporting threshold of 5 kg y$^{-1}$ is met for Hg and Hg containing
compounds that were manufactured, processed or otherwise used (includes by-products) or
contained in tailings and waste rock. For the AOSR domain, Hg emissions were updated in the
model from 2012 to 2015 using the NPRI point source Hg emissions data for each year. A summary
of Hg emissions from Athabasca oil sands upgrading facilities (NPRI) for 2012-2015 and temporal
trend from 2004-2017 are available in Table 1 and Figure 3, respectively. Based on NPRI, total
anthropogenic Hg emissions in Canada from the province of Alberta were 605 kg in 2015. Among
these, fossil fuel burning activities such as coal-fired power plants, waste incineration facilities
and other fossil fuel combustion contributed an estimated 221, 120 and 72 kg, respectively, which
represents 68% and, therefore, the bulk of total anthropogenic Hg emissions in Alberta. Iron and
steel production together with the cement industry (emitting 55 and 46 kg, respectively) contribute
another 14% and oil sands upgrading was a minor contributor (~ 25 kg) in 2015.

| Facility | Latitude | Longitude | 2012 | 2013 | 2014 | 2015 |
|---|---|---|---|---|---|---|
| Suncor Energy | 57.0033 | 111.466 | 35 | 37 | 0.439 | - |
| Syncrude - Mildred Lake | 57.0405 | 111.619 | 17 | 23 | 30 | 9.9 |
| Imperial Oil Resources - Cold Lake | 54.597 | 110.399 | 7 | 7.4 | 8.8 | 11 |
| Imperial Oil Resources - Kearl | 57.3969 | 111.071 | - | 1.1 | 4.3 | 4 |



| | | | | |
|---|---|---|---|---|
| **Sum of all four sources** | **59.0** | **68.5** | **43.5** | **24.9** |

Table 1: Athabasca Oil Sands Hg emissions (all in kg) reported to NPRI by oil sands processing
facilities, and used in the model. For the location of facilities in the AOSR see Figure 2.


Figure 3: Time series of total Hg emissions from oil sands processing facilities in the AOSR. Data
was compiled from the NPRI database. Numerical values and individual contributions from 2012-
2017 are available in Table 1.

**Model simulations**
Base model simulations at the three model simulation domains (i.e., global, NA and AOSR) were
performed using all sources of Hg emissions (as described earlier) and meteorological conditions
for the respective years from 2010 – 2015 to allow evaluation of modeled air concentrations with
measured air concentrations for all available years in the AOSR. Snowpack Hg measurements in
the AOSR started in 2012. Thus, the model-measurement comparison of snowpack Hg and the oil
sands Hg emissions impact study was conducted for the years 2012-2015.

Multiple controlled model simulations from 2012-2015 were performed choosing appropriate
geographic domains to assess the relative role of Athabasca oil sands Hg emissions on Hg burden
in the AOSR. The impact of Athabasca oil sands emissions was assessed by zeroing out emissions
of Hg from oil sands facilities in a controlled simulation using the AOSR domain. Contributions
of Hg emissions from biomass burning (in North America) and global anthropogenic sources to
the AOSR Hg levels were obtained by zeroing out emissions from these sources in controlled
simulations on North America and global model domains, respectively. Source apportionment of
the anthropogenic Hg deposition from worldwide sources was conducted using a series of global-



scale controlled simulations by zeroing out anthropogenic Hg emissions in different source
regions. In addition, controlled model simulations were performed to estimate the individual
influences of meteorology, biomass burning emissions and oil sands emissions on the interannual
variations in Hg deposition in the AOSR by successively adding these three temporal changes in
2013-2015.


**Mercury observations in the AOSR**
Simulated air concentrations and deposition of Hg were evaluated with observations of Hg in air
and snowpack in the AOSR. These measurements were recorded with instruments deployed for
air quality monitoring purposes and to study the atmospheric deposition of Hg species in the AOSR
(Parsons et al., 2013; Kirk et al. 2014; Gopalapillai et al., 2019). Air measurements were carried
out at three sites in the AOSR: Patricia-McInnes (2010-2018), Fort Mackay (2014-2018), and
Lower Camp (2012-2014). Measurements were made using Tekran 2537 Hg analysers for GEM,
and Tekran 1130/1135/2537 systems for speciated Hg (GOM and PBM) fitted with $PM_{2.5}$ and
$PM_{10}$ inlets (see map in Figure 2 for equipment placement and Figure 4-6 for data). Standard
operating procedures were provided by the Canadian Atmospheric Mercury Measurement
Network (CAMNet, (Steffen and Schroeder, 1999)). Air measurements of oxidized Hg
concentrations were carried out at only one site near Fort McKay in 2015 (Parsons et al., 2013).
Since Hg deposition to snow is mainly derived from the ambient oxidized Hg concentrations,
observations of snowpack Hg loadings provide additional constraint for modeled oxidized Hg
concentrations in air.

Snow samples were collected from 2012 to 2015 at 454 sites located at varying distances from the
major upgrading facilities (<1-231 km) to estimate total seasonal Hg loadings in surface snow in
the AOSR  (Gopalapillai et al., 2019; Kirk et al., 2014). Specifically, 90 (2012), 86 (2013), 140
(2014) and 138 (2015) samples were obtained from sites located close to the AOSR emission
sources (< 25 km) and at background sites further away from sources (> 120 km). Sample
collection was carried out in early to mid-March of each year at approximate maximum snowpack
depth based on Environment and Climate Change Canada's National Climate Data and
Information Archive historical snow accumulation data (GoC, 2019). Kirk et al. (2014) employed
ultra-clean handling and analysis protocols while taking care to avoid local contamination from



transportation since sites were accessed by helicopter and snowmobile. Mercury analysis in the
snow was carried out using cold vapour atomic fluorescence spectroscopy (Willis et al., 2018; Kirk
et al., 2014; EPA, 1996; Bloom and Crecelius, 1983). The determined snowpack Hg loading at the
end of the winter season represents lower limit of the net wintertime dry and wet deposition of Hg.
Hg deposited to snowpacks is partially reduced and re-volatilized to the air and lost during intra-
seasonal snowpack melting. Summertime measurements of Hg deposition by scavenging in rain
and direct uptake by vegetation, soils and waters were unavailable for model evaluation.

**Results and Discussion**
**Evaluation of model simulated mercury concentrations in air**
GEM-MACH-Hg has been extensively evaluated with comprehensive worldwide (including
Canada) observations, inter-compared with other Hg models, and applied to mercury assessments
in previous studies (Angot et al., 2016; Bieser et al., 2017; Dastoor et al., 2008; Dastoor and
Durnford, 2014; Durnford et al., 2010; Durnford et al., 2012; Fraser et al., 2018; Kos et al., 2013;
Travnikov et al., 2017; Zhou et al. 2021; AMAP 2013; CMSA, 2016; UNEP, 2018). Model
evaluation of ambient Hg in the AOSR is presented in this study. Figures 4-6 provide a comparison
of simulated (blue trace) and observed (red trace) daily averaged TGM concentrations in air at the
three observation sites (Figure 4: Patricia McInnis, 2010-2015; Figure 5: Lower Camp, 2012-2014;
and Figure 6: Fort McKay, 2014-2015), and how the model captured biomass burning events
(BBE) (green traces show modeled biomass burning contributions to TGM concentrations). While
some observations are incomplete (e.g., June 2013, Patricia McInnis), the data provide a detailed
picture of TGM surface concentrations near oil sands activities (see Figure 2 for details). In
general, data from all three observation sites and model simulation results agreed well with an
average squared Pearson correlation coefficient of 0.6, and measured and modeled median TGM
concentrations (± standard deviation) of 1.34±0.21 and 1.39±0.17 ng m$^{-3}$ (2011-2015) at Patricia
McInnis, 1.36±0.17 and 1.36±0.18 ng m$^{-3}$ (2013) at Lower Camp and 1.22±0.23 and 1.33±0.19
ng m$^{-3}$ (2014-2015) at Fort Mckay, respectively. The model captured the observed seasonal cycle
(typical in the northern hemisphere) with spring maxima and fall minima, shaped mainly by
surface fluxes of Hg such as the dominance of re-emission fluxes of Hg from snow in winter and
spring and uptake of Hg by vegetation in summer and fall. Transport of Hg from biomass burning
(i.e., wildfires) events in northern and western Canada yielded distinct Hg concentration peaks in



TGM concentrations in the AOSR (Figures 4-6). For 2011, biomass burning provided a large
contribution to overall TGM concentrations, which peaked during these events at Patricia McInnis;
however, no concurrent observations were available for the months of May and June. During the
large wildfire events in 2012 and 2015 (June-July), daily averaged TGM concentrations were
generally <2.5 ng m$^{-3}$, which were accurately reproduced by the model. However, as shown in
Figure 5 for the Lower Camp site in August 2013, there are discrepancies between modeled and
observed wildfire events. The impacts of biomass burning emissions on Hg burden in Canada and
the uncertainties in wildfire Hg emissions associated with the characterization of wildfire events
and emission levels using satellite and field data were described in a previous study (Fraser et al.,
2018). Low TGM concentration events in winter and early spring, such as those in March 2014 at
Patricia McInnis, were typically associated with clean air masses coming from the Arctic in AOSR.
Model-measurement agreement of TGM levels in the air is within the respective model and
measurement uncertainties and indicates that reported Hg emissions from AOSR facilities are
reasonable.

GOM and PBM observations were conducted at Fort McKay (a region dominated by natural boreal
forest) using PM$_{2.5}$ (captures particle sizes < 2.5 µm) and PM$_{10}$ (captures particle sizes < 10.0 µm)
inlets in AOSR for 2015, but significant measurement data gaps were present particularly in winter
and spring. Observed annual average concentrations were 1.02 ± 2.59 (GOM) and 3.47 ± 4.79 pg
m$^{-3}$ (PBM) using the PM$_{2.5}$ inlet, and 0.60 ± 1.11 (GOM) and 4.25 ± 8.23 pg m$^{-3}$ (PBM) using the
PM$_{10}$ inlet in 2015; these observations suggest a dominance of PBM in fine particles (< 2.5 µm)
at the Fort McKay site (17 km Northwest of AR6). The model simulated and observed average
TOM air concentrations and standard deviation (± 1σ) in 2015 were 4.74 ± 5.06 pg m$^{-3}$ and 5.74 ±
7.20 pg m$^{-3}$, respectively; observed data from both inlets was combined to reduce measurement
gaps. Episodes of high concentrations of particulate Hg (up to 72.9 pg m$^{-3}$), occurring
predominantly on coarse (> 2.5 µm) particles, , that were absent in the modeled PBM
concentrations were observed in March. The sources of coarse particles in the AOSR are currently
unknown, but fugitive dust from pet coke piles and roads as a result of oil sands mining activities
was suggested by Gopalapillai et al. 2019. It should be noted that uncertainty of a factor of 2 or
higher with oxidized Hg measurements has been reported (Kos et al., 2013; Gustin et al. 2015).
Comparable average GOM and PBM concentrations of 1.89 ± 8.31 and 3.82 ± 4.90 pg m$^{-3}$ (mean



± 1σ, 2009-2011), respectively, have been measured at a site 8 km from a coal-fired power plant
in Genesee, AB (about 500 km southwest of Fort McMurray). Seasonal cycles at the two sites
(Fort McKay and Genesee) were similar, with TOM maxima in May-June. Since Hg deposition to
snow is primarily driven by the uptake of ambient oxidized Hg species in snowfall and snowpack,
the robustness of model simulated oxidized Hg in air was further tested by comparing modeled
snowpack Hg loadings with measurements (see next section).

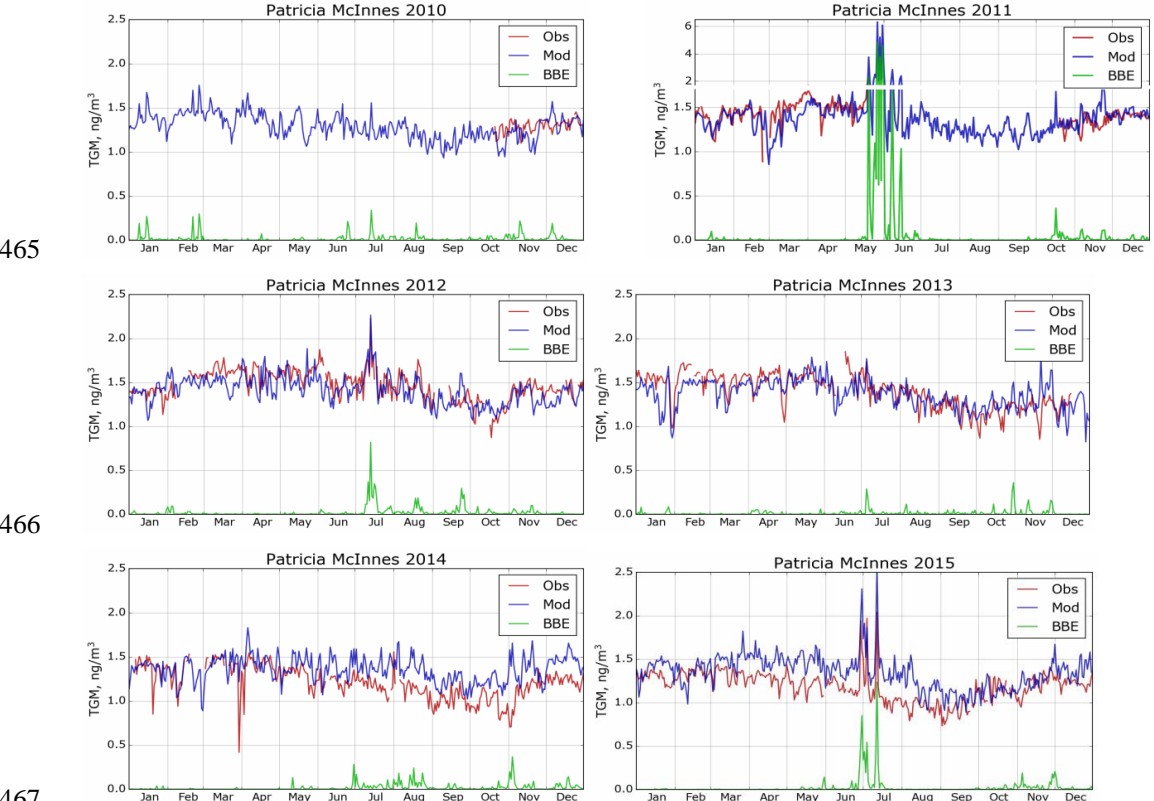

Figure 4: Simulated and observed daily averaged surface air TGM concentrations in AOSR for the
site Patricia-McInnes (2010—2015). Obs – observations; Mod – model estimation; BBE –
modeled biomass burning contributions. Note the larger range of the y-axis to plot the strong
biomass burning event in May and June of 2011.




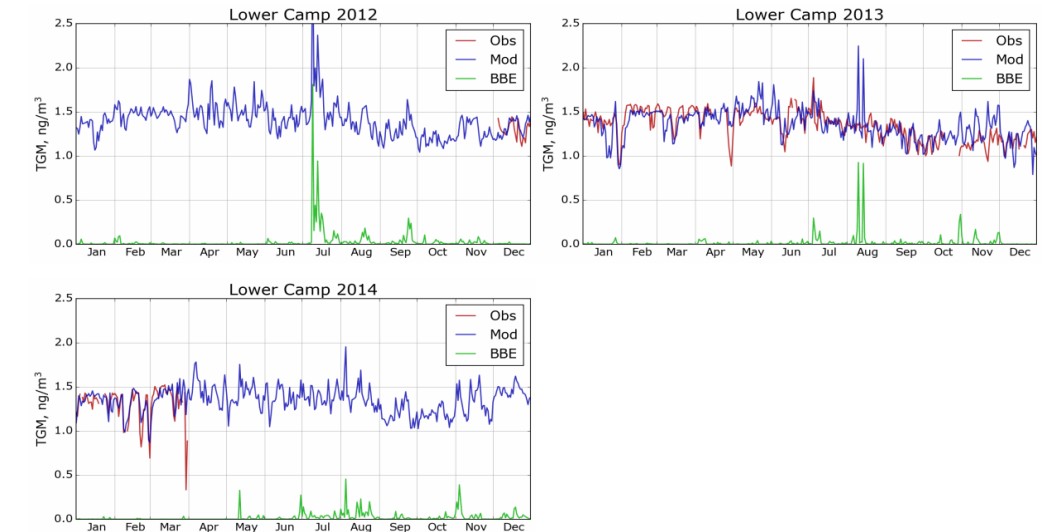


Figure 5: Simulated and observed surface air TGM concentrations in AOSR for the site Lower
Camp (2012—2014). Obs – observations; Mod – model estimation; BBE – modeled biomass
burning contribution.


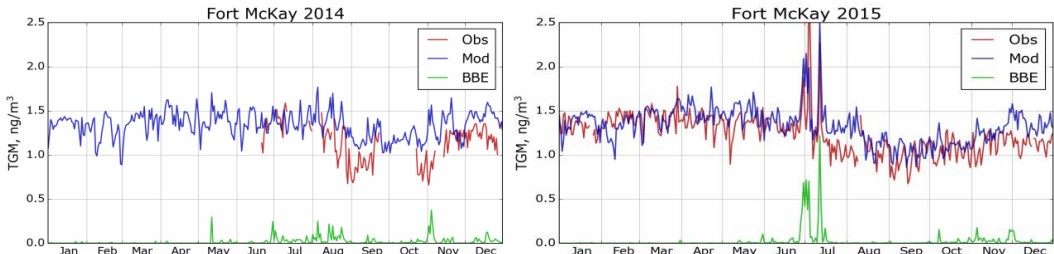


Figure 6: Simulated and observed surface air TGM concentrations in AOSR for the site Fort
McKay (2014 and 2015). Obs – observations; Mod – model estimation; BBE – modeled biomass
burning contributions.


For the purpose of comparing ambient GEM concentrations in the AOSR with other Canadian
regions, Figure 7 provides a map of modeled annual average surface air Hg concentrations of GEM
for Canada in 2013. In general, model estimated surface air GEM concentrations agreed well with
available observations (in circles), including western Canada, the Pacific coast, and the AOSR.
There is a general gradient in GEM concentrations from higher concentrations in the west (1.5 ng
m$^{-3}$) to lower concentrations in the east (1.3 ng m$^{-3}$). The average air concentrations of GEM in

the AOSR (1.40 ng m⁻³, 2012-2015) reflected the background GEM levels in Canada. The
simulated large-scale pattern in GEM concentrations is consistent with, and reflects, a dominant
role of trans-Pacific transport of GEM from East Asian Hg sources into Canada and the high
Arctic. GEM concentrations are slightly higher in major urban centres and regions of current and
past anthropogenic activities such as energy production from coal-fired power plants and mining.
The hotspot in Figure 7 near the Saskatchewan/Manitoba border is the former copper-zinc smelter
near Flin-Flon, MB, which ceased operations in 2010 (Ma et al., 2012). The soils in the
surrounding region remain heavily contaminated with Hg. The re-emission of accumulated legacy
mercury in soils (Eckley et al., 2013) is responsible for the highly elevated GEM concentrations
in air..

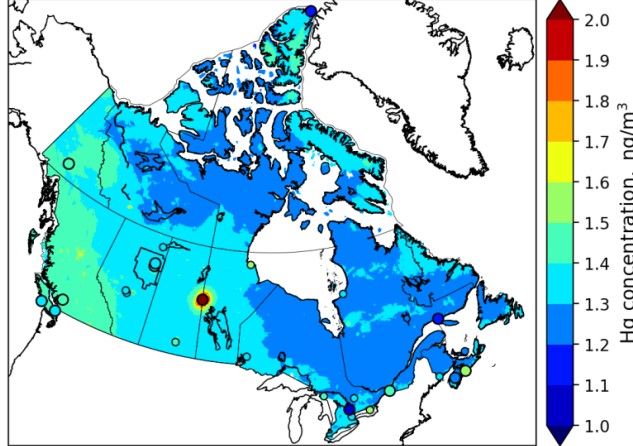


Figure 7: Model simulated spatial distribution of annual average surface air GEM concentrations
in Canada in 2013; colors in circles show observed concentrations for 2013 (large circles) and
previous years (small circles).

**Evaluation of model simulated mercury accumulation in snow**
Figure 8 compares total Hg loadings in snow simulated by the model with observations (in circles)
at the end of winter for years 2012-2015 in the AOSR. Cooke et al., (2017) used dated lake
sediment cores to reconstruct deposition trends and anthropogenic enrichment in the region, but
several correction factors needed to be applied to estimate Hg deposition fluxes and only two lakes
were cored in the direct vicinity of oil sands operations. By comparison, seasonal snowpack Hg
data provide the distribution of net total Hg deposition in the region with a large number of





sampling sites a short distance (< 25 km) away from sources. However, it should be noted that Hg
deposition in the snow is partially reduced and reemitted as well as adsorbed in surface soils due
to diffusion and intra-seasonal melt; therefore, snowpack Hg represents the lower limit of net
wintertime deposition. Observations at the sampling sites close to sources had the highest
snowpack Hg loadings with decreasing concentrations as one moves further away from the
immediate source region; the same spatial pattern was predicted by the model, and is most evident
for the years with the largest emissions (2012 and 2013; Figure 8). Snow Hg contents at the
background sites in the Peace Athabasca Delta region in the north were significantly lower, which
was also well reproduced by the model. The figure shows high spatiotemporal variability in snow
Hg loadings, which are related to changes in meteorological factors as well as oil sands emissions
(as discussed later). The decline in both snowfall amounts and oil sands emissions led to lower
snow Hg loadings in 2014 and 2015. Figure 9 shows the model simulated average snow depths in
the AOSR and the observed depths at the Mildred Lake site close to the Syncrude upgrader. The
model simulates snow amounts and interannual variations accurately. The model-estimated
seasonal snow accumulations were 62, 183, 104 and 71 cm between October to May in 2012, 2013,
2014 and 2015, respectively.  An intense intra-seasonal melting event at the end of February was
predicted by the model in each year, which is inline with observations. The largest melting event
occurred in 2015, which caused over half of the snow accumulation to melt, and, thus, loss of half
of seasonal snowpack Hg loadings. Modeled snow Hg loadings are in agreement with  Gopalapillai
et al. (2019), who reported a temporal decrease in snow Hg loadings near-field (< 8 km from AR6),
from an average load of 510 ng/m$^2$ in 2008 to 175 ng/m$^2$ in 2016. Relative importance of inter-
annual changes in meteorological conditions and oil sands emissions to wintertime Hg deposition
is discussed a later section.

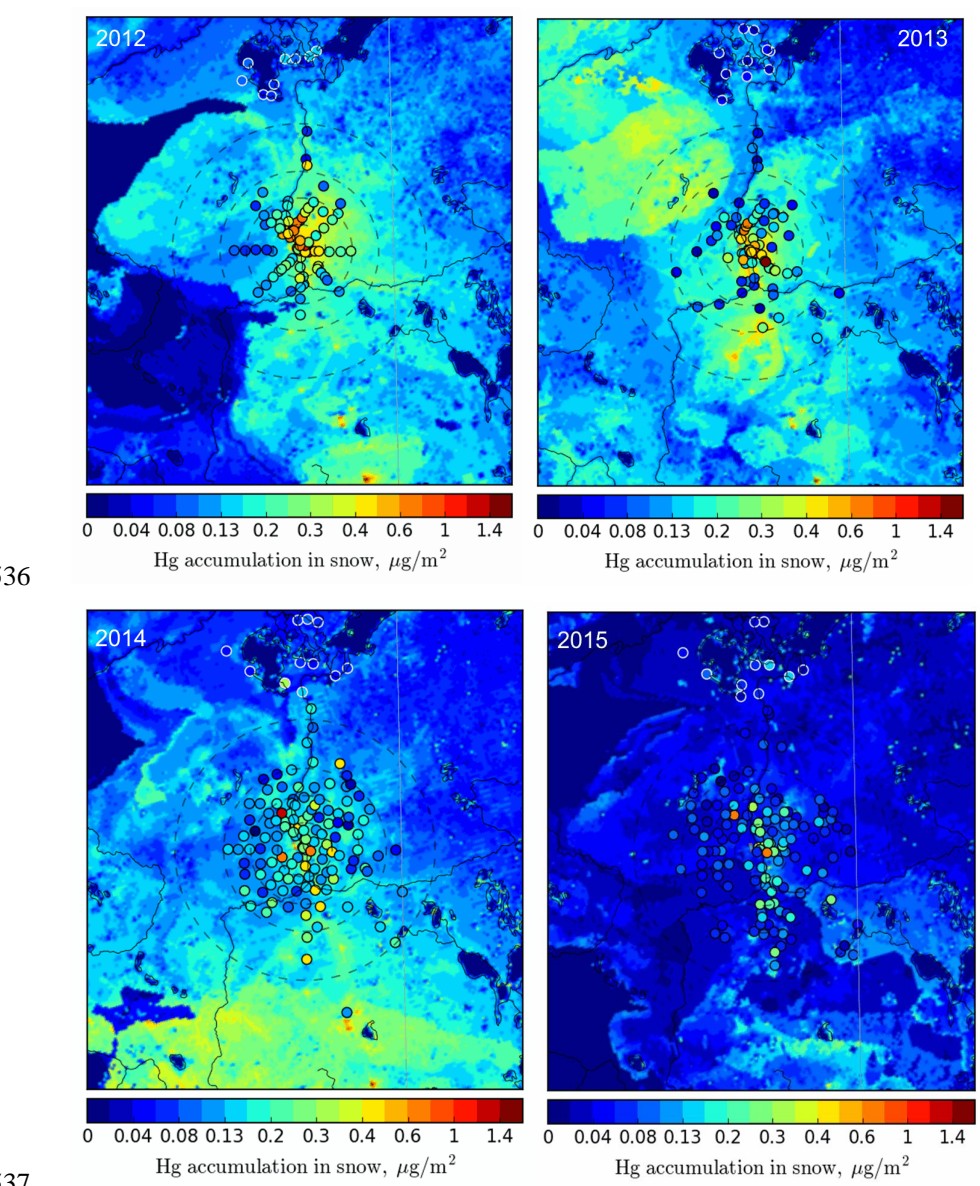



Figure 8: Seasonally accumulated Hg loadings in snow in AOSR from 2012 to 2015: modeled
(background map) and observed values (colors in circles). Circles radii: 25, 50, 75, 120 km.



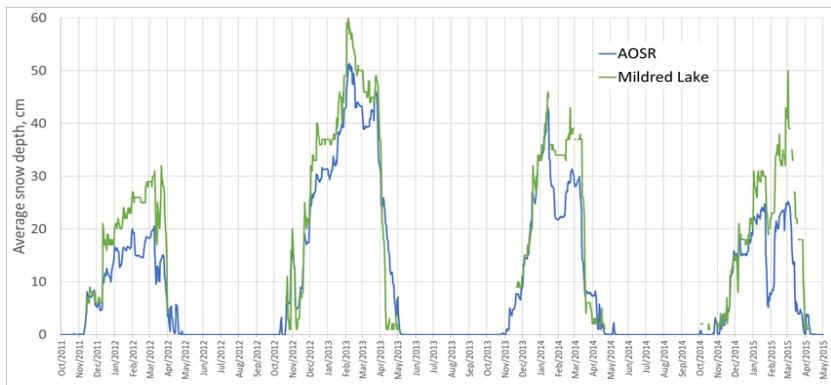


Figure 9: Daily averaged model simulated (blue) and observed snow depths (green) (cm) in 2012-
2015 in the AOSR. Modeled values are averaged over the entire AOSR domain and the observation
site is Mildred Lake, Alberta, a few km east of the Syncrude oil sands upgrader.

Figure 10 compares average modeled and observed snow Hg loadings at the sampling locations
within 25 km, 25-50 km, 50-75 km, 75-120 and > 120 km distances from AR6.  Inter-annual
changes in meteorology and oil sands emissions led to decreases in total Hg loads from 0.52±0.21
to 0.22±0.09 µg m$^{-2}$ within 25 km of AR6 (from 2012 to 2015) in the snowpack for observation
and from 0.39±0.21 to 0.08±0.06 µg m$^{-2}$ for model estimates sampled at sites.  The snow Hg
loadings of up to 0.7 µg m$^{-2}$ were simulated by the model in the immediate vicinity of Hg emitting
sources for 2012 (Figure 8). Emitted amounts of Hg from oil sands facilities were reported to the
NPRI with the caveat that not all emissions, e.g., emissions of mercury that are part of fugitive
dust releases, are captured by the inventory. Brief episodes of Hg on larger particles (2.5-10 µm
size) were observed at Fort McKay in late winter, likely originating from fugitive dust in the
AOSR. These possible sources of Hg emissions and related deposition (in the vicinity of sources)
were not included in the model. At > 120 km from AR6, snowpack loadings were very low for all
years at < 0.1 µg m$^{-2}$ with small inter-annual variability, and indicate background Hg
concentrations at this distance.

While the strong decrease away from the source is mirrored in Figure 10 for the years 2012 and
2013 (dropping from about 0.4 µg m$^{-2}$ at sites located <25 km from AR6 to < 0.1 µg m$^{-2}$ at sites >
120 km away), the weaker signature from Figure 8 for the years 2014 and 2015 is more clearly
represented in Figure 10, consistent with declines in reported oil sands emissions (see Table 1 and





Figure 3). Modeled snow Hg loadings closer to the oil sands sources were lower compared to
observed values in 2015. A sensitivity model simulation was conducted for 2015 by replacing
NPRI reported Hg emissions from oil sands facilities in 2015 with 2014 values. The sensitivity
model simulation matched the observed Hg loadings in the snow in 2015 at all distances; these
results suggest that either NPRI Hg emissions from oil sands facilities were slightly under-
represented or there was an unaccounted area source (such as from fugitive dust) of Hg in 2015.

Model estimates and observations agreed well for all distances evaluated, and demonstrate the
model's ability in correctly simulating the impacts of changes in Hg emissions and
physicochemical processes in the cryosphere. The high variability in the observed snowpack data
within 50 km of AR6 indicates that there are likely other local sources around mining facilities
that impact local deposition (such as fugitive dust from coke pile and roads). However, modeled
estimates at sampling locations agreed with observed snow Hg loadings within one standard
deviation, and suggest that unaccounted sources of Hg do not have a significant impact on
deposition in the AOSR, likely due to their episodic nature as suggested by observed ambient
concentrations of particle-bound mercury.



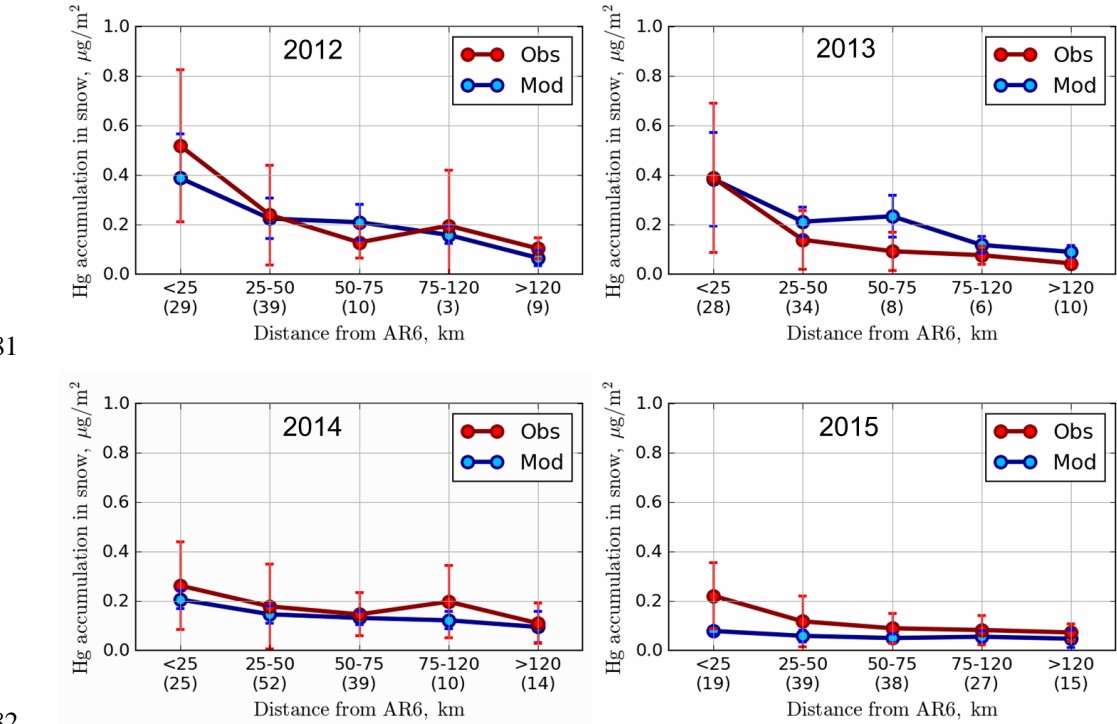



Figure 10: Average modeled (µg m⁻²; blue) and observed (µg m⁻²; red) end of winter Hg loadings
in snowpack within 25 km, 25-50 km, 50-75 km and 75-120 and > 120 km distances from AR6
along with ±one standard deviations. Modeled accumulated Hg in the snow was sampled at the
observation sites. Numbers in parentheses provide number of observation sampling sites in each
distance cluster.







Comparison of modeled annual wet and total deposition (wet plus dry deposition) fluxes of Hg in
the AOSR with other locations in Canada is presented in Figure 11 for 2013. In general, spatial
distributions of wet and total deposition fluxes followed patterns of precipitation (high in the east,
south and mountainous regions of Canada), industrial activities (high in southern Canada),
vegetation density (boreal and temperate forests) as well as Hg transport from the US (higher in
the east). Figure 11 shows good agreement with observed wet deposition fluxes (noted in circles)
in coastal (Saturna Island, BC), rural (Southern Alberta) and urban areas (Egbert, ON). While
direct measurements of annual total deposition fluxes are not available, the distribution of Hg
deposition fluxes in Canada was found to be consistent with Canada-wide lake sediment inferred


deposition fluxes (Muir et al. 2009). Average annual total deposition fluxes in the AOSR were
16.9, 15.7, 18.3 and 17.5 µg m$^{-2}$ in 2012, 2013, 2014 and 2015, respectively, slightly higher than
in the other regions of northern Alberta (~14 µg m$^{-2}$/y) and lower than average Hg deposition flux
in southern Alberta (~25 µg m$^{-2}$/y). The highest deposition up to 80 µg m$^{-2}$ occurred in southern
Ontario in Canada due to the presence of local anthropogenic mercury emissions in these regions.

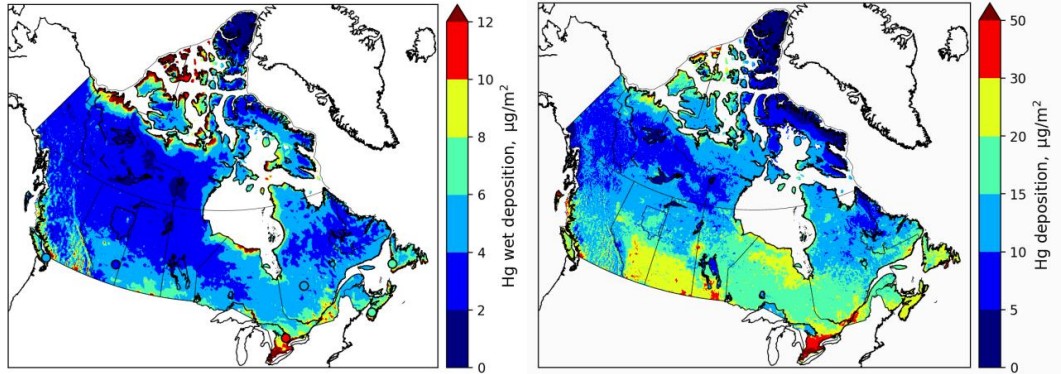


Figure 11: Model simulated and observed annual Hg wet deposition for 2013 (left) (colors in
circles show observed wet deposition for 2013) and simulated annual total Hg deposition (right)
(wet plus dry deposition) in Canada for 2013.

**Impacts of oil sands developments and wildfires on mercury levels in air and deposition**
Employing GEM-MACH-Hg, the impacts of Hg emissions from oil sands developments in the
AOSR on surface air concentrations of Hg species (i.e., GEM and TOM), snowpack Hg loadings,
and annual Hg deposition were investigated for the years 2012-2015. Since Northwest Canada is
a region of high wildfire activity (Fraser et al. 2018), the relative role of Hg emissions from
biomass burning in North America on the Hg burden in the AOSR was also examined.
Figures 12 & 13 provide spatial distributions of simulated annual average surface air
concentrations of GEM (globally transported and the dominant ambient Hg species) and TOM
(regionally transported and efficiently deposited Hg species) (left panels) for the years 2012 to
2015 along with their contributions (as % increases) from oils sands emissions (OSE, middle
panels) and biomass burning emissions (BBE, right panels) in the AOSR and the surrounding
region. GEM air concentrations were 1.4 ng m$^{-3}$ in the AOSR in 2012-2015, which is within the


range of GEM concentrations observed in Alberta (i.e., 1.2-1.5 ng m$^{-3}$ in 2012). While annual average GEM concentrations were slightly elevated close to the major upgraders (> 1.5 within 5 km vs 1.4 ng m$^{-3}$ 200 km away from AR6) in the AOSR, GEM concentrations were found to be elevated up to 1.8 ng m$^{-3}$ in surrounding regions of the AOSR due to local wildfires in 2012-2015. Since the lifetime of GEM in the air is between 0.5-1 year, GEM concentrations are largely driven by global transport in the AOSR (and Canada) with only minor contributions from local emissions. Oil sands emissions increased atmospheric GEM concentrations up to 2.3% in 2012 and 2013, and negligibly (up to 0.9%) in low OSE years 2014-2015, only very close to the upgraders (i.e., within 2.5 km). Wildfire activities are highly variable from year to year, and can significantly impact GEM concentrations in the AOSR in summertime (Fraser et al. 2018). Biomass burning contributed to 1.0-2.2% increases in average GEM concentrations in and around the AOSR (Figure 12, right panels), making biomass burning a more important source of GEM than OSE in the region. Strong regional biomass burning events led to large increases in GEM concentrations of up to 35% (2012-2015) in the AOSR and the surrounding regions.

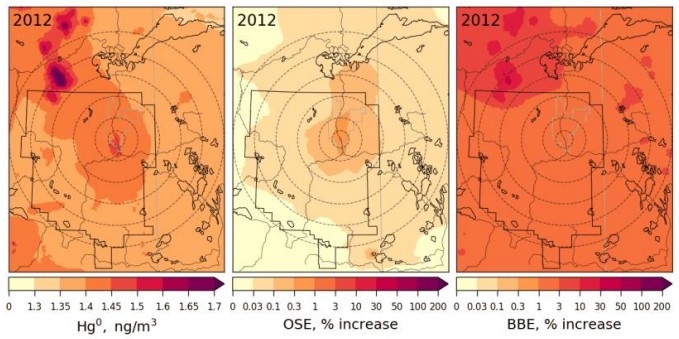


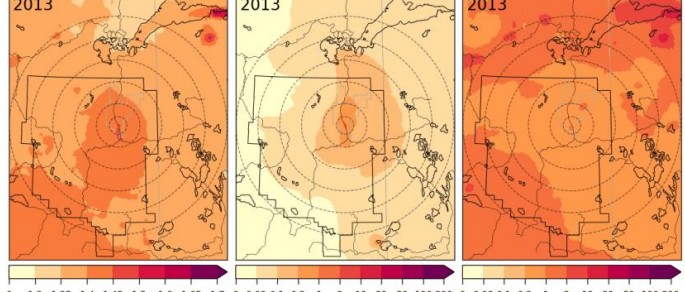




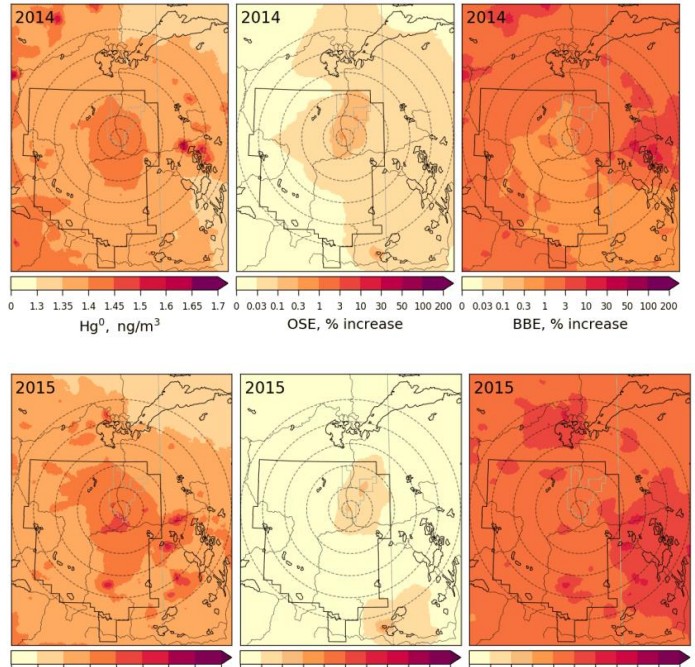



Figure 12: Annual average surface air concentration of GEM (left) and concentration enrichments (%) due to Hg emissions from Athabasca oil sands operations (OSE, middle) and biomass burning in North America (BBE, right) for the years 2012 to 2015. The AOSR is marked as an approximate rectangle, and concentric distance circles are at 20, 50, 100, 150, 200 and 250 km from AR6.

While average surface air TOM concentrations in the AOSR were only 3.3 pg m$^{-3}$ (consistent with observations), hot spots were modelled in the immediate vicinity of the major upgraders (> 25 pg m$^{-3}$ within 5 km from AR6 in 2012-2013) in the AOSR (Figure 13, left panels). In 2014-2015, TOM concentrations around AR6 were about half of 2012-2013 (12 pg m$^{-3}$), consistent with reported changes in Hg emissions from the respective facilities. OSE are found to be the main and a major contributor of oxidized Hg concentrations in surface air close to oil sands sources, increasing background concentrations over 30% within 100 km and 60% within 50 km from AR6 in 2012-2013, particularly in the northeast sector of the AOSR. Wildfire emissions played a minor role in ambient TOM concentrations in the region, contributing to < 1% increases in 2012, 2013 and 2015, but increased to ~6% in 2014 as a result of higher wildfire activites. Hg emitted from




oil sands operations as oxidized species is deposited efficiently by precipitation and uptake from
terrestrial surfaces in the vicinity of the sources. By comparison, most of the GEM emissions are
transported out of the region except for a small fraction being deposited locally via direct
vegetation uptake and conversion to oxidized species and dry deposition. Oxidized Hg species
emitted from global sources do not reach the AOSR via long-range transport due to their short-
lived nature. As a result, OSE-related Hg deposition in the AOSR consists primarily of TOM,
whereas, long-range transport of GEM accounts for the deposition in the AOSR attributed to
outside sources.  Wildfire emissions are mostly assumed to be emitted as GEM as indicated by
observations (Friedli et al. 2001).

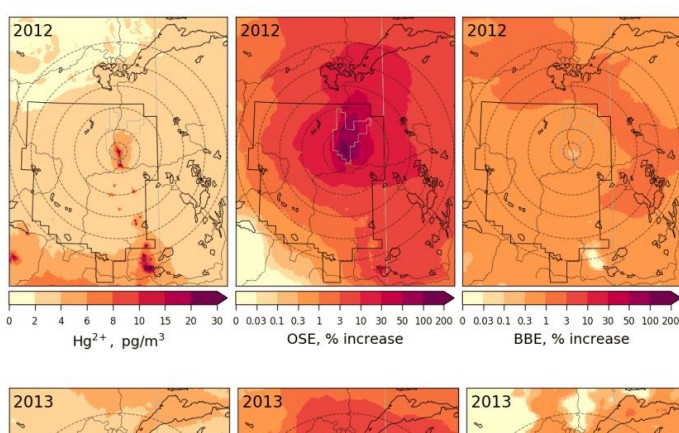


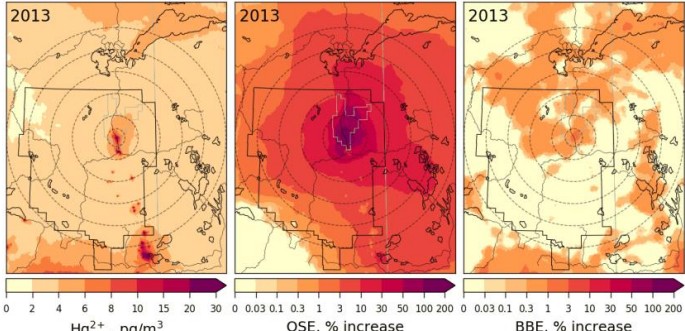






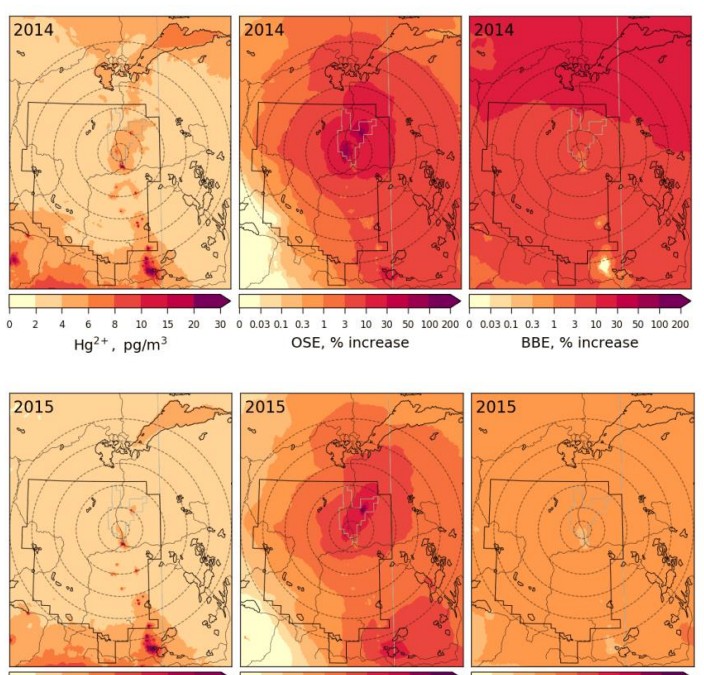



Figure 13: Annual average surface air concentration of TOM (sum of GOM and PBM, left), and concentration enrichments (%) due to Hg emissions from Athabasca oil sands operation (OSE, middle) and biomass burning in North America (BBE, right) for the years 2012-2015. AOSR is marked as an approximate rectangle and concentric distance circles are at 20, 50, 100, 150, 200 and 250 km from AR6.

Figures 14 and 15 provide spatial distributions of modelled annual total mercury deposition (Figure 14, left panels) and seasonally accumulated Hg loadings in the snow (Figure 15, left panels), and their source attributions to OSE (Figure 14, middle panels; Figure 15, right panels) and BBE (Figure 14, right panels) in the AOSR in 2012-2015. Mercury deposition fluxes from 7-28 $\mu$g m$^{-2}$y$^{-1}$ (15.6 -18.3 $\mu$g m$^{-2}$y$^{-1}$, averages) were modelled in the AOSR in 2012-2015, originating from all Hg emission sources - global primary and legacy anthropogenic and geogenic (including oil sands and biomass burning) emissions. Since the contribution of global transport of GEM to the ambient total Hg concentrations in the AOSR is much larger than the contributions of OSE and BBE (Figure 12) and GEM concentrations are typically 2-3 order of magnitude higher than TOM





concentrations (which have higher contributions from OSE, Figure 13), deposition of imported
GEM makes up a major portion of the annual Hg deposition in the AOSR on a broad spatial scale,
despite its lower Hg deposition efficiencies than TOM (Figure 14). Similar to ambient TOM
concentrations, modelling reveals the impact of OSE to Hg deposition to be greatest in the vicinity
of upgraders, i.e., average increases of 17%, 20%, 8%, and 3% within 20 km of AR6 in 2012,
2013, 2014 and 2015, respectively, and < 1 % beyond 50 km in all years. Model results reveal a
larger impact of OSE on Hg deposition in the regions northeast of oil sands sources, consistent
with observations and prevailing wind direction and speed (Kirk et al. 2014). Average Hg
deposition contributions due to BBE (increases of 1.4-13% ) were higher than OSE contributions
(increases of 0.3-1.3%) across 200 km of oil sands operations in 2012-2015. Wildfires in the
region led to localized increases in Hg deposition of up to 193% and 101% in 2012 and 2014,
especially northwest of the AOSR. Mercury emissions from electricity generation in southern
Alberta accounted for a general decrease in Hg deposition fluxes from south to north around the
AOSR.


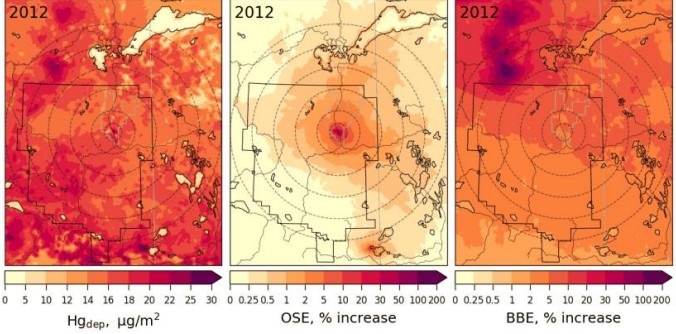


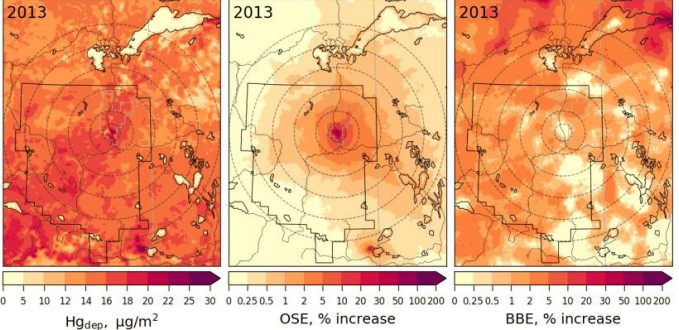



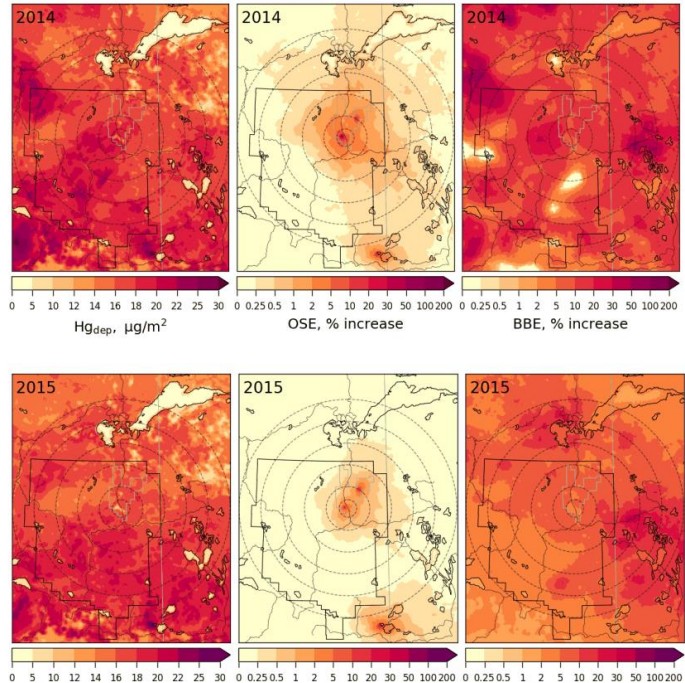



Figure 14: Annual total Hg deposition flux (left) and deposition enrichments (%) due to Hg emissions from Athabasca oil sands operations (OSE, middle) and biomass burning in North America (BBE, right) in 2012-2015. The AOSR is marked as an approximate rectangle and concentric distance circles are at 20, 50, 100, 150, 200 and 250 km from AR6.

Snowpack Hg accumulations from the start of the snow season to the end of winter (roughly coinciding with the maximum snow accumulation period) and their contributions from oil sands Hg emissions were estimated for 2012-2015 (Figure 15). Background snow Hg loadings (without the impact of OSE, middle panels) were spatially highly variable (up to 1.4 µg m$^{-2}$) in the region between 2012-2015. The higher snow Hg background levels resulted from both the regional transport of Hg from southern Alberta as well as spatial inhomogeneity in the accumulation of snow. Closer to OSE sources, total Hg loadings in snow reached up to 1.0 µg m$^{-2}$ ($< 20$ km from AR6) in 2012-2014 (Figure 15). In 2015, emissions from oil sands-related activities were the lowest and total Hg loadings corresponded to background emissions. The impact of OSE was notably greater to the snowpack Hg loadings, including the spatial extent, than to the annual Hg





deposition (Figure 15, right panels). Average increases of 55%, 43%, 35% and 7% in snow Hg
amounts were simulated within 50 km of AR6 in 2012, 2013, 2014 and 2015, respectively, as a
result of OSE. Regions northeast of the AOSR showed increases of 27-44% in snow Hg levels in
2012 and 2013 and 3-24% in 2014 and 2015 between 50-100 km from AR6. Model results support
the conclusions of previous studies that oil sands Hg emissions have a large impact on snow Hg
loadings near the oil sands emission sources with decreasing contributions away from AR6 (Kelly
et al., 2010; Kirk et al., 2014). The distinctive pattern of higher snow Hg loadings in the northeast
region surrounding the AOSR was also reported (Kirk et al., 2014). Model results reveal high
spatiotemporal variability in background snow Hg loadings; this is related to variability in snowfall
amounts, meteorological conditions affecting melting and snowpack Hg processes including
redox, air-snow exchange and transport to soils.

Average annual Hg deposition fluxes in the AOSR were 13.3 (2015) to 18.5 (2013) $\mu g\ m^{-2}y^{-1}$
within 10 km, 15.0 (2015) to 16.9 (2013) $\mu g\ m^{-2}y^{-1}$ between 10-20 km, and ~ 16 $\mu g\ m^{-2}y^{-1}$ 50 km
away from the major oil sands emission sources. In the AOSR, winter (and snow cover) can last
up to six months (from November to April) with maximum snow depths in January-February.
Winter (November-April) and summer (June-August) periods contributed to ~20% and 50%,
respectively, of annual Hg deposition in AOSR. In Figure 16, three representative months in the
winter (December to February) and summer (June to August) seasons, each, are chosen to present
the inter-seasonal contrast in OSE impacts on Hg deposition along with the impact on annual
deposition as a function of distance from AR6.

Seasonally, OSE accounted for the largest Hg deposition increases in winter months: ~230-500%
(2013), 146-374% (2012), 94-104% (2014) and 40-43% (2015) within 10 km; 75% (2013), 57%
(2012), 25% (2014) and 5% (2015) at 20 km; and 24-33% (2012-2013) and 6-12% (2014-2015) at
50 km distance from the major oil sands upgraders. In summertime, lower deposition increases
due to OSE were estimated, ~13-56% (2012-2013) and 3-7% (2014-2015) within 10 km, and <
7% (2012-2015) at 20 km from AR6. Annually, OSE accounted for deposition increases of ~24-
70% (2012-2013), 14% (2014) and < 5% (2015) within 10 km, 10% (2012-2013) and < 5% (2014-
2015) at 20 km, and < 4% (2012-2015) at 50 km from the major oil sands emission sources. These
seasonal variations are consistent with inter-seasonal differences in Hg deposition pathways (i.e.





the dominant role of GEM uptake by vegetation in summer from global sources, and uptake of
local TOM emissions by snowfall and snowpack as the main pathway in wintertime deposition)
(Graydon et al., 2006; Obrist et al., 2016; Zhang et al. 2009). The influence of OSE to summertime
and annual depositions is also more limited spatially (up to 30 km of OSE) than to wintertime
deposition (up to 100 km of OSE), consistent with observations (Kirk et al, 2014; Gopalapillai et
al., 2019).


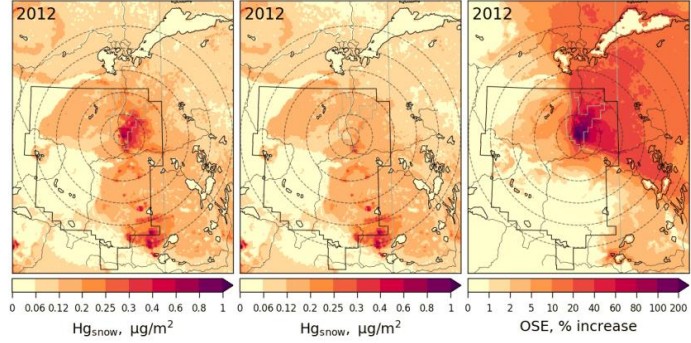
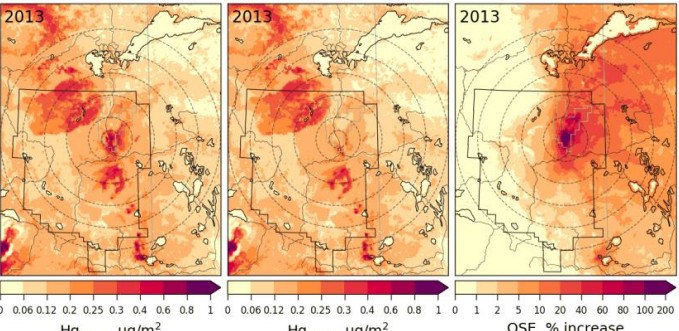

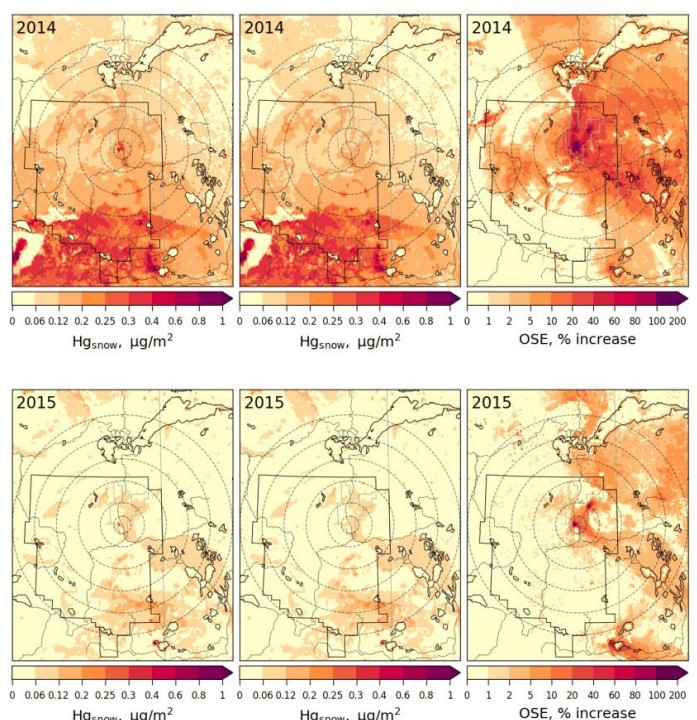



Figure 15: Seasonally accumulated Hg loadings in snow with all Hg emissions (left) and without Athabasca oil sands Hg emissions (middle), and enrichments (%) in seasonally accumulated Hg loadings in snow due to Athabasca oil sands Hg emissions (OSE, right) in 2012-2015. The AOSR is marked as an approximate rectangle and concentric distance circles are at 20, 50, 100, 150, 200 and 250 km from AR6.





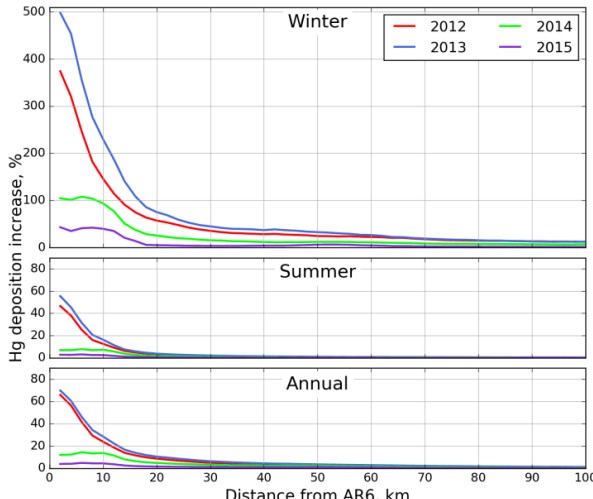

Figure 16: Average Hg deposition enrichments (%) due to Athabasca oil sands emissions in winter

from December to February (top), in summer from June to August (middle), and annually (bottom)

for 2012 (red), 2013 (blue), 2014 (green) and 2015 (pink) by distance from AR6.

**Process attribution of interannual variations in mercury deposition**

The interannual differences noticed in Figure 16 raises the question of the contributing factors to

the interannual variability of Hg deposition in different seasons, especially close to the processing

facilities (i.e., within a 10 and 20 km radius). The relative importance of variations in

meteorological conditions and changes in OSE and BBE on the temporal changes in Hg deposition

fluxes from 2012 to 2015 were analyzed. Since meteorological changes are expected to occur

regardless of changes in emissions, a controlled model simulation was first conducted by applying

meteorological changes only from 2012 to 2015. Subsequently, two additional model simulations

were performed by including changes in BBE and OSE successively. It should be noted that

interannual variations in meteorological factors (synoptic as well as local scale) affect overall

seasonal and annual deposition rates and, therefore, the magnitude of the impacts of various

emissions on the deposition, irrespective of the changes in emissions; thus, the results presented

here are cumulative contributions of changes in meteorology and emissions. Figure 17 presents

process attribution of interannual changes in winter (top), summer (middle) and annual (bottom)

Hg deposition rates from 2012-2015 within 0-10 km (left) and 10-20 km (right) from AR6.. The

lower panels illustrate Hg deposition source contributions from global emissions (green; global





anthropogenic (except oil sands), geogenic and re-emission), OSE(red) and BBE(purple), and the
upper panels show process contributions of changes in meteorology (blue), oil sands (red) and
biomass burning (purple) emissions to interannual changes in total Hg deposition.

While wintertime Hg deposition fluxes were relatively low (2.6-3.6 µg m$^{-2}$, November-April; 0.3
– 0.8 µg m$^{-2}$, December-February) in the AOSR,  oil sands emissions were a major source of Hg
deposition close to the oil sands sources as explained earlier, contributing to 70-80% of deposition
within 10 km of AR6 in high oil sands emission years (2012 and 2013). Wintertime (net) Hg
deposition to northern landscapes is controlled by cryospheric processes, which exhibit strong
interannual variations; therefore, interannual variation in wintertime Hg deposition is strongly
controlled by meteorological conditions including snowfall amounts, wind speed, surface air
temperature, solar insolation, and intra-seasonal melting affecting air-snow-soils exchange
processes of mercury (Faïn et al., 2013).   In 2015, a large snowmelt event at the end of February
effectively removed about half of the accumulated mercury in snow resulting in much lower snow
Hg content at the time of sampling (see Figure 9).

Surface temperature and intra-seasonal melting have a large impact on how much of the deposited
Hg in the snow is re-emitted back to the atmosphere and how much is adsorbed to surface soils,
altering snow Hg loadings and net wintertime Hg deposition. Since 2013 experienced deeper
snowpack and less inter-seasonal melting, a larger fraction of snowpack Hg was reduced and
revolatilized, leading to a lower net Hg deposition despite slightly higher oil sands Hg emissions
compared to 2012. Conversely, lower snowpack depth and a strong melting event at the end of
February in 2015 allowed a large fraction of snowpack Hg to be transferred and retained in
underlying soils increasing net Hg deposition, particularly the background deposition contribution.

Within 10 km of major oil sands sources, wintertime variations in meteorology led to Hg
deposition declines of 17% in 2013 and 2014 and increases of 10% in 2015 along with OSE-led
deposition declines of 10% (2013), 35% (2014) and 56% (2015).When combined, the net effect of
these two factors were overall reductions in wintertime Hg deposition fluxes of 27% (2013), 52%
(2014) and 46% (2015),  relative to 2012.At a distance of 10-20 km from the oil sands sources,
changes in meteorology led to a 54% increase in wintertime Hg deposition in 2015, but the overall


deposition only increased by 19%, because the decline in oil sands Hg emissions reduced the
deposition by 35%. River discharge rates and Hg concentrations are reported to be highest in the
spring meltwater flood (between 3 ng/L and 16 ng/L, up from typically <2 ng/L at their lowest
annual level) in tributaries of the Athabasca River and pose risk to the downstream environments
(Kelly et al., 2010; Wasiuta et al., 2019). Since the ground is still frozen at the time of spring
freshet, Hg runoff is derived from seasonal snowpack loadings and mobilization of Hg from
surface soils, both of which are contaminated by oil sands emissions in proximity of the sources
and show a sensitivity to changes in Hg emissions from oil sands developments.

Compared to winter, AOSR summertime background Hg deposition fluxes were significantly
higher (~6.3-7.5 $\mu g\ m^{-2}$, 2012-2015) and less variable in space and time, and OSE contributions to
total deposition were relatively lower (~ 0.05-0.5 $\mu g\ m^{-2}$ within 10 km and 0.01-0.2 $\mu g\ m^{-2}$ from
10-20 km, 2012-2015). In addition, summertime biomass burning emissions contributed to Hg
deposition of 0.1-0.4 $\mu g\ m^{-2}$ (2012-2015). Summertime Hg deposition to terrestrial systems is
temporally less variable than wintertime deposition as it is predominantly driven by Hg uptake by
vegetation and soils followed by wet deposition. Changes in oil sands emissions played a more
significant role than the meteorological factors in summertime inter-annual Hg deposition
variations.. Compared to 2012, changes in meteorology, biomass burning and oil sand emissions,
respectively, led to changes in summertime Hg deposition fluxes by -3%, -2%, and +7% in 2013,
+3%, +2% and -15% in 2014, and -1%, +4% and -20% in 2015, resulting in overall changes in Hg
deposition by +2% (2013), -10% (2014) and -17% (2015), within 10 km of major oil sands sources.
Interannual variations in precipitation amounts and its impact on the wet deposition of Hg was the
primary reason for the meteorology-related changes in summertime Hg deposition fluxes.

Since summertime deposition contributes to about half of the annual deposition, interannual
changes and their responsible factors in annual Hg deposition fluxes had a similar pattern as
summer, with a relatively larger impact of changes in OSE on Hg deposition fluxes in the
immediate vicinity of oil sands sources. Relative to 2012, deposition increases were 6 (2014) and
1% (2015) due to variations in meteorology and 2% (2014-2015) due to biomass burning, and
deposition declines were 15 (2014) and 23% (2015) due to reduction in oil sands Hg emissions.
This results in overall reductions in annual Hg depositions of 7 (2014) and 20% (2015) within 10





km of AR6. These model results demonstrate that reduction in Hg emisisons from oil sands

processing activities lead to measurable declines in mercury deposition fluxes in AOSR. Further

away from sources (right panel, Figure 17), the changes in meteorology and oil sands emissions

resulted in comparable changes in Hg deposition rates (+9 (2014) and +5 % (2015), meteorology;

-4 (2014) and -9% (2015), OSE) along with 3(2014) and 2(2015)% increases in deposition due to

BBE, resulting in relatively smaller overall changes (+8% (2014) and -2% (2015)) in Hg deposition

fluxes. Interestingly, land clearing in the AOSR contributes to reduced background Hg deposition

fluxes due to the reduction in foliage Hg uptake; average background Hg deposition fluxes were

about 1 $\mu$g m$^{-2}$ lower within 10 km as compared to Hg deposition fluxes 20 km away from the

major oil sands activities.

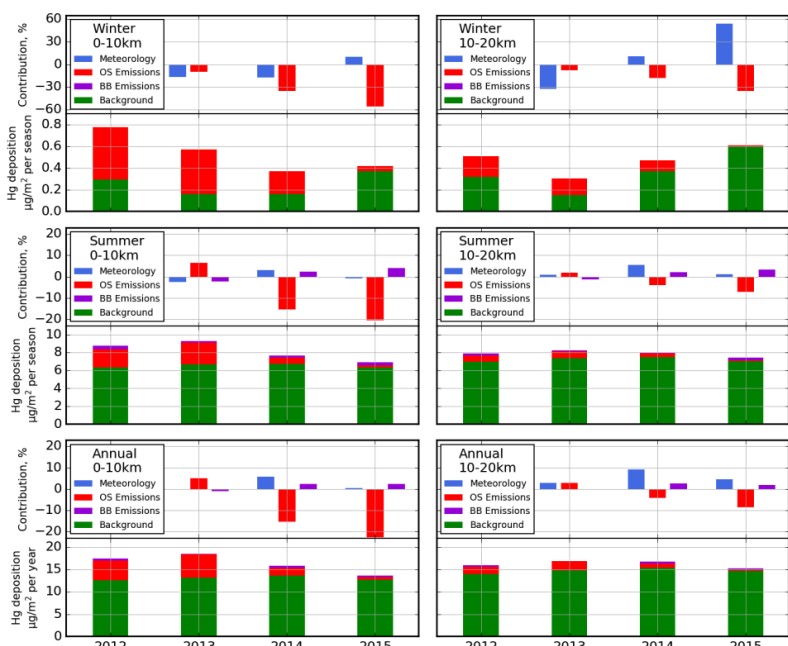

Figure 17: (a)December – February, (b) June – August and (c) yearly averaged source

apportionment of total Hg depositions (lower panels) in 2012-2015, and contributions of changes

in meteorology, Athabasca oil sands emissions and biomass burning emissions (only in summer)

(top panels) to the changes in total Hg depositions in 2013-2015 relative to 2012, within 10 km

(left plot) & 10-20 km (right plot) of AR6.



**Source apportionment of the background mercury deposition**

As noticed in Figure 14-16, background Hg (long-range transport from global source regions; excludes impact of oil sands emissions, but includes impact of all other Hg emissions in Canada) is responsible for the majority of annual Hg deposition in the AOSR (except in winter in the vicinity of major oil sands Hg emission sources). The average annual background Hg deposition in the AOSR was 15.3-16.7 µg m$^{-2}$y$^{-1}$ in 2012-2015. This includes ~40% deposition from contemporary global anthropogenic Hg emissions (excluding Hg emissions from Athabasca oils sands activities) and ~60% from global geogenic emissions and re-emissions of legacy mercury deposition (of both anthropogenic and geogenic origin). The model was applied to investigate the relative proportions of background anthropogenic Hg deposition fluxes contributed from various worldwide emission source regions, including Canada, in the AOSR (Figure 18). Almost 50% of the background anthropogenic Hg deposition originated from East and Southeast Asia, a region of high economic activity and high energy demand, which is sourced for the most part by coal-fired power plants. The model estimated that foreign anthropogenic sources accounted for over 98% of the background anthropogenic Hg deposition in the AOSR of which present-day emissions in East Asia, Southeast Asia, South Asia, Sub-Saharan Africa, Europe and the United States contributed to approximately 38%, 9%, 8%, 8%, 7%, and 2%, respectively. Emissions from present-day anthropogenic sources in Canada (excluding oil sands sources in AOSR) contributed to < 2% of the background anthropogenic Hg deposition nationally including the AOSR. In proximity of oil sands activities, oil sand Hg emisons are a significant source of Hg deposition as demonstrated earlier in this study. By comparison, oil sands developments currently have a negligible impact on Hg deposition on a broader spatial scale in Canada.  These results highlight the need for worldwide mitigation efforts, in addition to the local efforts, to reduce the risks of mercury contamination in the AOSR.



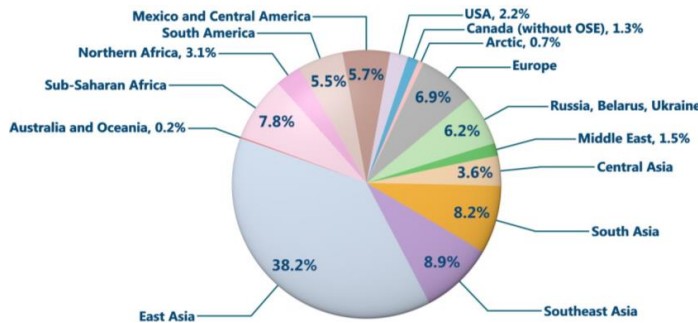


Figure 18: Deposition contributions from global anthropogenic source regions (excluding
Athabasca oil sands Hg emissions) to the average contemporary anthropogenic Hg deposition
portion (40% of total deposition) of the total deposition in Athabasca Oil Sands Region in 2015.

**Conclusions**
An assessment of mercury levels in air and deposition in the Athabasca oil sands region (AOSR)
in Northern Alberta, Canada, was conducted to investigate the contribution of Hg emitted from oil
sands activities on the surrounding landscape using a 3D process-based Hg model in 2012-2015.
The model-simulated Hg burden in the region was first evaluated with multi-year observations of
air concentrations of Hg and seasonally accumulated Hg in snow. Model-measurement agreement
of Hg surface air concentrations and snow loadings in AOSR were within the measurement and
modeling uncertainties and implies that NPRI reported emissions of Hg from oil sands operations
(i.e., 59, 69, 44 and 25 kg in 2012, 2013, 2014 and 2015, respectively) are consistent with Hg
burden in the region. Air concentrations of Hg(0) in the AOSR (1.4 ng m$^{-3}$) were at a similar level
as found in Northern Alberta, and were within the range of concentrations in Canada (1.2-1.6 ng
m$^{-3}$). Background Hg(0) concentrations in Canada are dominated by long-range transport, with a
slightly larger impact in the west, and, thus, contribution of oil sands activities to Hg(0)
concentrations in AOSR was minimal (< 0.1%, average enrichment). During the summer season,
Hg emissions originating from regional wildfires were found to be an episodically important
source of atmospheric Hg(0), with daily averaged concentrations peaking to 2.5 ng m$^{-3}$ (Parsons
et al. 2013; Fraser et al., 2018). Average total oxidized Hg concentrations (gaseous plus
particulate) in the air were elevated above background by 55% and 65% in 2012 and 2013,
respectively, and over 10% in 2015 within 50 km of upgrading facilities (particularly in the
northeast sector) in the AOSR as a result of oil sands emissions.






The level and spatial extent of the impact of oil sands emissions to winter, summer and annual Hg
deposition fluxes were examined in high (2012-2013) and low (2014-2015) oil sands Hg emission
years. In 2012-2015, annual average total Hg deposition fluxes of 15.6-18.3 µg m$^{-2}$y$^{-1}$) were
simulated in AOSR with deposition in winter (November-April) and summer (June-August)
contributing to 20% and 50%, respectively. The emission sources of Hg deposition in the AOSR
are global anthropogenic (including Canadian emissions), natural and reemissions of legacy Hg
deposition (including biomass burning emissions). Similar to other regions in Canada, on a broader
scale, Hg deposition in the AOSR is dominated by mercury transported from global sources, with
a small (and highly spatiotemporal variable) impact from regional biomass burning events. In
proximity to oil sands sources, however, total Hg deposition in wintertime was largely driven by
oils sands emissions.  Deposition increases of up to 146-500% occurred within 10 km of oil sands
sources in the high emission years 2012 and 2013; summertime and annual Hg deposition increases
due to oil sands emissions were 13-56% and 24-70%, respectively, within 10 km of sources for
the same years. In lower oil sands emission years (2014 and 2015), Hg deposition increases due to
oil sands activities declined to 40-104% in winter and 5-14% annually within 10 km of oil sands
sources. At 20 km from the oil sands operations, oil sands-related Hg deposition enhancements
were not as large, with increases of 57-75% in winter, and 10% annually in 2012 and 2013. The
spatial extent of the OSE influence on Hg deposition was also greater in winter relative to summer
(~100 km vs 30 km from major Hg emitting facilities).

Finally, factors contributing to the inter-annual variations (i.e., changes in meteorological
conditions, oil sands emissions and wildfire emissions) in seasonal and annual Hg deposition
fluxes and relative source attributions in AOSR were examined from 2012 to 2015. Wintertime
(net) Hg deposition to northern landscapes is controlled by Hg deposition to snowpacks by direct
uptake and via snowfall and post-depositional processes, which exhibit strong inter-annual
variations. Relative to 2012, while changes in meteorological conditions led to a reduction in
wintertime Hg net deposition fluxes by ~ 17% in 2013-2014, and an increase by 10% in 2015
within 10 km of oil sands sources, changes in oil sands emissions led to deposition reductions of
10%, 35% and 56% in 2013, 2014 and 2015, respectively, resulting in an overall reduction in
wintertime Hg depositions of 27%, 52% and 46% in 2013, 2014 and 2015, respectively.



Gopalapillai et al. (2019) reported temporal decline in snowpack total Hg loadings near-field, from
an average load of 510 to 175 ng/m$^2$ from 2008 to 2016. At a distance of 10-20 km from the oil
sands sources, while changes in meteorology led to a 54% increase in wintertime deposition in
2015 relative to 2012, the decline in oil sands emissions led to a reduction in the deposition by
35%, resulting in an overall increase in Hg deposition of 19%. Summertime Hg deposition to
terrestrial systems is temporally less variable than wintertime deposition as it is predominantly
driven by Hg uptake by vegetation and soils, and by wet deposition; thus, changes in oil sands
emissions played a more significant role in summertime inter-annual variations in Hg deposition
than the meteorological factors. Compared to 2012, changes in meteorology, biomass burning and
oil sand emissions led to changes in summertime deposition by -3%, -2%, and +7% in 2013, +3%,
+2% and -15% in 2014, and -1%, +4% and -20% in 2015, resulting in overall changes in Hg
deposition by +2%, -10% and -17% in 2013, 2014 and 2015,  respectively, within 10 km of major
oil sands sources. On an annual basis, in 2014 and 2015, variations in meteorology and biomass
burning emissions led to deposition increases of 1-6% and 2%, respectively, and reduction in oil
sands Hg emissions led to declines between 15-22%, resulting in an overall reduction in annual
Hg deposition of 7-20% within 10 km of AR6. In 2015, at 10-20 km away from sources, Hg
deposition increase due to changes in meteorology plus biomass burning was approximately equal
to deposition decline due to changes in oil sands emissions, resulting in smaller (<8%) changes in
Hg deposition fluxes.

Oil sands Hg emissions are found to be important sources of Hg contamination to the local
landscape in proximity of the processing activities, particularly in wintertime. Although Hg
deposition is higher in summertime (mainly driven by long-range transport), oil sands Hg
emissions contribute to a notably higher proportion of deposition in wintertime in the AOSR. Thus,
the impact of oil sands emissions is more easily detected in snow Hg observations (Kirk et al.,
2014). Wintertime Hg deposition rates are also more influenced by interannual changes in
meteorological conditions compared to summer. Regarding the environmental importance of
seasonal Hg deposition, it is likely that a major portion of summertime deposition remains bound
to vegetation and subsequently transferred to soils, where it can be partially sequestered and partly
reemitted back to air or mobilized in aquatic systems on long timescales of decades to centuries
(Zhou et al. 2021). In contrast, wintertime deposition (and partially summertime wet deposition)





can be transferred to the local aquatic system via runoff more readily (i.e, on an annual time scale).
Model findings reveal that year-to-year changes in meteorological conditions not only significantly
influence the rate of Hg deposition but, additionally, can either exacerbate or diminish the impact
of changes in oil sands emissions on Hg deposition, particularly in winter. Thus, meteorological
changes can confound the interpretation of trends in short-term monitoring data. In addition,
meteorological changes related to climate change can influence the deposition trends. Accurate
reporting of point and area Hg emisions related to oil sands activities,  long-term monitoring of
Hg in air and terrestrial ecosystems, and the application of process-based Hg models are crucial to
understanding systematic changes in Hg levels and their causes in the AOSR.

**Acknowledgements**
We thank our ECCC colleagues Paul Makar, Sandro Leonardelli and Stewart Cober and in the
Pollutant Inventory and Reporting Division for their insightful comments and careful internal
review of the manuscript. This project was supported by the Joint Oil Sands Monitoring (JOSM)
program of ECCC.

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
