# Peer review of "Impact of Athabasca oil sands operations on mercury levels in air and deposition"

_Atmospheric Chemistry and Physics, 2021_

## Author Response (AR1)

**Responses to comments on the manuscript acp-2021-296**

Dear Editor,

We have carefully responded to the reviewers' comments (in blue) and have revised the manuscript accordingly.

Thank you very much for your consideration of our manuscript for publication in ACP.

Best Regards,
Ashu Dastoor

**RC1**: Anonymous Referee #1, 17 May 2021

**General comments:**

The authors present a detailed study of the impact of Hg emissions in 2012 – 2015 from oil sand processing in the Athabasca Oil Sands Region in Alberta on the mercury deposition in the region and the larger surroundings. After a validation of the model by a comparison with available measurements, the authors analyse the impact of the decreasing emissions, the role of biomass burning and meteorological conditions, as well as the contributions of regional and global Hg sources.

The paper is generally well ordered and written. It reads well despite its length. The abstract is too detailed and long for an abstract and I recommend to shorten it by presenting the crucial results only qualitatively. After that I recommend the publication of the manuscript after removing some spelling problems.

We thank the reviewer for positive and constructive comments on our paper. We have shortened the abstract in the revised manuscript as suggested.

**Specific comments:**

Line 318: Possibility

Corrected

Line 341: (all in kg yr$^{-1}$)

Corrected

Lines 423-425: Higher GEM oxidation rate in summer and the resulting maximum of Hg wet deposition in summer is probably the more important process.

Our model sensitivity analysis to various Hg processes shows that oxidation has comparatively small impact on the seasonal cycle of GEM. GEM bi-directional surface fluxes have a much larger impact on shaping the GEM seasonal cycle (see Zhou et al. 2021), except in regions of high oxidation rates such as the Arctic in springtime. We have added citation to Zhou et al. 2021 in the revised manuscript.

Zhou, J., Obrist, D., Dastoor, A., Jiskra, M. & Ryjkov, A. Vegetation uptake of mercury and impacts on global cycling. Nat. Rev. Earth Environ. 2, 269-284 (2021).

Line 454: "pet coke piles"?

This expression is correct.

Line 887: "emissions"

Corrected

**RC2**: Anonymous Referee #2, 08 Jul 2021

**General comments:**

The manuscript presents detailed analysis of mercury pollution levels in the Athabasca oil sands region in Northern Alberta, Canada. The study involves both measurement data on mercury concentration in air and seasonal accumulation in snow as well as multi-scale simulations with a chemistry transport model. In particular, the study analyses the impact of mercury emissions from oil sands developments and biomass burning of mercury concentration and deposition in the region. Additionally, contribution of global, regional and local sources to mercury deposition levels in the region is investigated and major processes responsible for the inter-annual variation of pollution levels are analyzed.

The subject of the manuscript is relevant to the scope of the journal and the work makes up a new and original contribution. The methodology used is adequate and explicitly stated. The manuscript will be suitable for publication after addressing the comments mentioned below.

We thank the reviewer for positive and constructive comments on our paper.

**Specific comments:**

Figure 1: "…*The Athabasca Oil Sands Region is indicated with an approximate rectangular shape within northeastern Alberta, bordering Saskatchewan*."

The "approximate rectangular" is very poorly seen in the figure as well as in all other figures of the manuscript.

We have revised the concerned figures to make the oil sands region more visible by making the window blue with a thicker line.

Lines 615-619: "*... spatial distributions of simulated annual average surface air concentrations of GEM ... and TOM ... along with their contributions (as % increases) from oils sands emissions (OSE, middle panels) and biomass burning emissions (BBE, right panels)...*"

The concentration/deposition increase (in %) due to OSE and BBE is among the key characteristics analyzed in the manuscript and mentioned in the conclusions. However, it is not clearly defined in the text. More certain definition is needed to understand particular numbers and figures given in the text.

Both oil sands and biomass burning Hg emissions (in the context of total global and regional emissions) and the design of model simulations conducted to study their impacts are clearly defined in detail in preceding sections including introduction, objectives, "the model and emissions inputs", and "model simulations". We think that this is sufficient and additional definition is not needed in this section.

Lines 789-790, Figure 17: "*... the upper panels show process contributions of changes in meteorology (blue), oil sands (red) and biomass burning (purple) emissions to interannual changes in total Hg deposition.*"

Similarly, it is not clear how the relative contributions of particular processes to deposition changes were calculated. More detailed description is needed.

The model simulations for estimating process contributions are already described in the section "model simulations" and in the first paragraph of the section mentioned by the reviewer (please see lines 776-785).

To improve the clarity, we have revised the lines 776-785 as follows: *Since meteorological changes are expected to occur regardless of changes in emissions, a controlled model simulation was first conducted by applying only meteorological changes from 2012 to 2015. Subsequently, two additional model simulations were performed by successively adding BBE and OSE changes from 2013-2015. The differences in these simulations provided the relative process contributions. It should be noted that, in addition to the changes in emissions, the BBE and OSE impacts on Hg deposition also depend on changes in meteorological conditions (synoptic as well as local scale), thus the results presented here are cumulative contributions of changes in meteorology and emissions.*

Lines 903-907: "*... Model-measurement agreement of Hg surface air concentrations and snow loadings in AOSR ... implies that NPRI reported emissions of Hg from oil sands operations ... are consistent with Hg burden in the region.*"

It seems to be too strong conclusion repeated in the Abstract taking into account that contribution of the AOSR region emissions to GEM air concentration is negligible and it does not exceed 55% for mercury accumulation in snow.

This conclusion is based on 4 years of model simulations during which oil sands emissions were changing. In the vicinity of oil sands activities, oxidized Hg concentrations in air and snow Hg loadings are highly sensitive to oil sands Hg emissions. For example, within 10 km of oil sands sources, average enhancement in wintertime deposition to snow due to oil sands emissions were 250 – 350% in 2012-2013, but only 50-100% in 2014-2015, because of declining oil sands Hg emissions (see Figure 16). Since year to year variations in modeled oxidized Hg and snow Hg levels in the vicinity of oil sands facilities follow the changes in oil sands emissions quite consistently, also seen in observations, this conclusion is accurate.

*Conclusions, Abstract:* Overall, the conclusions and abstract seem to be too extensive and are overloaded with plenty of numerical details. In my view, their shortening would improve readability of the manuscript.

We have shortened the abstract as suggested by the reviewer. We have retained the numerical details in the conclusion for easy accessibility of results to the readers.